# MLLM Is a Strong Reranker: Advancing Multimodal Retrieval-augmented Generation via Knowledge-enhanced Reranking and Noise-injected Training

## Abstract

Multimodal Large Language Models (MLLMs) have demonstrated remarkable capabilities in processing and generating content across multiple data modalities. However, a significant drawback of MLLMs is their reliance on static training data, leading to outdated information and limited contextual awareness. This static nature hampers their ability to provide accurate and up-to-date responses, particularly in dynamic or rapidly evolving contexts. Though integrating Multimodal Retrieval-augmented Generation (Multimodal RAG) offers a promising solution, the system would inevitably encounter the multi-granularity noisy correspondence (MNC) problem, which hinders accurate retrieval and generation. In this work, we propose **RagVL**, a novel framework with knowledge-enhanced reranking and noise-injected training, to address these limitations. We instruction-tune the MLLM with a simple yet effective instruction template to induce its ranking ability and serve it as a reranker to precisely filter the top-k retrieved images. For generation, we inject visual noise during training at the data and token levels to enhance the generator's robustness. Extensive experiments on four datasets verify the effectiveness of our method. Code and models are available at https://anonymous.4open.science/r/RagVL-F694.

## 1 Introduction

As an attempt towards Artificial General Intelligence (AGI), Large Language Models (LLMs) have made significant strides in language understanding and human-like text generation (Brown et al., 2020; Achiam et al., 2023; Touvron et al., 2023). However, true AGI requires more than just linguistic capabilities. It necessitates a comprehensive understanding and interaction with the world, encompassing multiple modalities beyond text. Thus, the recent progress of Multimodal Large Language Models (MLLMs) in handling multimodal information has attracted the community. By processing and generating content across different modalities, MLLMs aim to create a more holistic and nuanced understanding of the world, closer to how humans perceive and interpret information. This integration of modalities enables MLLMs to perform tasks that require contextual understanding from multiple data sources, such as Visual Question Answering (VQA) (Goyal et al., 2017; Hudson & Manning, 2019; Marino et al., 2019), Table Question Answering (Lu et al., 2022), Text-to-image Generation (Ramesh et al., 2021; Yu et al., 2022; Aghajanyan et al., 2022), etc.

Nevertheless, the promising performance of language models primarily relies on the knowledge implicitly stored in their massive parameters, leading to several issues such as long-tail knowledge gaps (Asai et al., 2024), generating hallucinations (Ye & Durrett, 2022), and poor model interpretability. To better adapt to knowledge-intensive tasks and real-world scenarios, Retrieval-augmented Language Models (RALM) (Lewis et al., 2020; Lin et al., 2023; Izacard & Grave, 2020; Karpukhin et al., 2020) employ a dense retriever to retrieve up-to-date knowledge from external memories for grounded generation. Similarly, Multimodal Retrieval-augmented Generation (Multimodal RAG) enhances MLLMs by dynamically retrieving relevant information from external multimodal data sources before generation. This allows the models to incorporate real-time, contextually accurate visual information, significantly improving the factuality and accuracy of their outputs.

Figure 1: Difference between traditional VQA and multimodal knowledge-seeking question answering. An example from WebQA (Chang et al., 2022) reveals the challenge of multi-granularity noisy correspondence (MNC).

As illustrated in Figure 1, to answer the information-seeking query, the model must retrieve and reason over external visual knowledge, which differs from traditional VQA on the left and is non-trivial. To solve this, MuRAG (Chen et al., 2022) makes the first endeavor to extend RAG to multiple modalities. It is built upon ViT (Dosovitskiy et al., 2020) and T5 (Raffel et al., 2020) and pre-trained to encode image-text pairs for both answer generation and retrieval. MuRAG embeds items into an external memory and handles queries for retrieving multimodal knowledge from the same memory.

However, integrating multimodal RAG would inevitably introduce the multi-granularity noisy correspondence problem (MNC) (Huang et al., 2021). As shown in Figure 1, MNC refers to the noise at two different granularities: (I) *Coarse-grained noise (query-caption)*. During the retrieval stage, coarse-grained captions result in retrieving similar but negative images (e.g., *'Uxmal Gobernador Uxmal,Yucatan, Mexico Governer's Palace, seen from House of the Old Woman'* and *'Palacio del Gobernador-Uxmal-Yucatan-Mexico0277 Palace of the Governor in Uxmal'*). (II) *Fine-grained noise (query-image)*. The retriever and generator must distinguish fine-grained visual elements to formulate the responses. Any discrepancies between the images and the question can introduce noise, thereby compromising the accuracy of the results. In this scenario, the classical CLIP (Radford et al., 2021) struggles to match the query with the image during the retrieval phase (see in Table 1). Also, identifying the correct correspondence amidst the fine-grained noise to provide an answer to the query is a challenge.

To this end, we propose **RagVL**, a novel framework with knowledge-enhanced reranking and noise-injected training, to mitigate MNC in multimodal RAG. In the retrieval stage, we instruction-tune the MLLM with a simple yet effective instruction template to induce its ranking ability. Given that MLLMs are inherently capable of understanding cross-modal information, we employ the fine-tuned model as a reranker to evaluate the relevance between the query and the image, which precisely selects top-$N$ candidates that are more related to the query semantically. Subsequently, we apply an adaptive threshold to filter the candidates, collaborating with the reranker to alleviate the fine-grained mismatches. To further mitigate the impact of fine-grained mismatches during the generation phase, we introduce noise at both data and token levels in the training process. Specifically, at the data level, we perform negative sampling for single-image input questions within the single/multiple-image interleaved dataset, supplementing them with references from hard negative images. At the token level, we introduce additional visual uncertainty to images through Gaussian noise and reassign training loss weights by comparing the logits of the distorted and original inputs.

In a nutshell, the main contributions of this work are as follows:

- We achieve effective and robust multimodal retrieval-augmented generation with a three-stage pipeline. Additionally, we address the inherent multi-granularity noisy correspondence (MNC) problem in multimodal retrieval-augmented generation.

- We introduce the knowledge-enhanced reranking and noise-injected training technique to mitigate the coarse-grained and fine-grained noise from MNC.

- Extensive experiments on multimodal knowledge-seeking QA and retrieval tasks demonstrate the effectiveness of the proposed framework.

## 2  RELATED WORK

### 2.1  MULTIMODAL LARGE LANGUAGE MODEL

Recent advances in Multimodal Large Language Models (MLLMs) have demonstrated impressive performances in handling multi-format information (Driess et al., 2023; Huang et al., 2024; Achiam et al., 2023). MLLMs are generally built upon existing Large Language Models (LLMs) and integrating visual information as input tokens by utilizing an additional vision encoder (*e.g.* CLIP) and a bridging connector (*e.g.* MLP). For instance, LLaVA (Liu et al., 2024b;a) adopts one/two linear MLP to project visual tokens and align the feature dimension with word embeddings, while BLIP-2 (Li et al., 2023) leverages a group of learnable query tokens to extract information in a query-based manner. By connecting the visual and textual modalities, MLLMs significantly enhance human-AI interaction and demonstrate remarkable capabilities in understanding and generating multimodal content. Despite these advances, MLLMs tend to underperform in knowledge-intensive tasks (*e.g.* WebQA and MultimodalQA (Talmor et al., 2021)) that require seeking up-to-date information. Since the knowledge stored in their massive parameters is currently limited, it is crucial for MLLMs to resort to external memories for grounded generation.

### 2.2  MULTIMODAL RETRIEVAL-AUGMENTED GENERATION

Enhancing language models by incorporating relevant information from diverse knowledge sources has been shown to improve performance across various NLP tasks (Borgeaud et al., 2022; Lewis et al., 2020). DPR (Karpukhin et al., 2020) trains the retriever using in-batch documents and samples negative examples for contrastive learning, allowing the pre-trained retriever to excel in open-domain question answering. REALM (Guu et al., 2020) and RAG (Lewis et al., 2020) treat the retrieved passages as latent variables and train the retriever-generator system jointly, leading to more effective retrieval-augmented generation models. Inspired by textual RAG, Plug-and-play (Tiong et al., 2022) retrieves relevant image patches using GradCAM (Selvaraju et al., 2017) to localize relevant parts based on the query. MuRAG (Chen et al., 2022) proposes the first multimodal retrieval-augmented Transformer, which accesses an external non-parametric multimodal memory to augment language generation. To better connect candidates and model their relations during retrieval, SKURG (Yang et al., 2023) employs an Entity-centered Fusion Encoder to align sources from different modalities and determines the number of retrieval steps adaptively using a unified Retrieval-generation Decoder. However, none of these works specifically focus on MNC in multimodal RAG, which is the primary focus of our research. Experimental results show that the proposed knowledge-enhanced reranking and noise-injected training effectively improves multimodal RAG.

## 3  METHODOLOGY

### 3.1  PRELIMINARIES

The traditional Retrieval-augmented Language Model (RALM) acquires knowledge from the external memory $\mathcal{M}$ and utilizes the knowledge in grounded outputs to promote accurate and explainable generation. The retriever $\mathcal{R}$ first retrieves the top-$K$ most relevant contexts $\mathcal{C} = \{c_1, \cdots, c_k\}$ from $\mathcal{M}$ for the given question $q$. Subsequently, the autoregressive language model generates answers based on these retrieved contexts. Under the multimodal setting, the retriever needs to compare the textual queries with the multimodal documents and find the best matches for the generator $\mathcal{G}$. In this paper, we focus on retrieving the visual-related contexts to study open-world multimodal question answering.

### 3.2  MULTIMODAL RETRIEVER

We follow the dual-encoder architecture based on CLIP text encoder $\Phi_{text}$ and image encoder $\Phi_{img}$. Before the retrieval stage, given image-query pairs $(v, q)$ from the dataset $\mathcal{D}$, we first apply the image

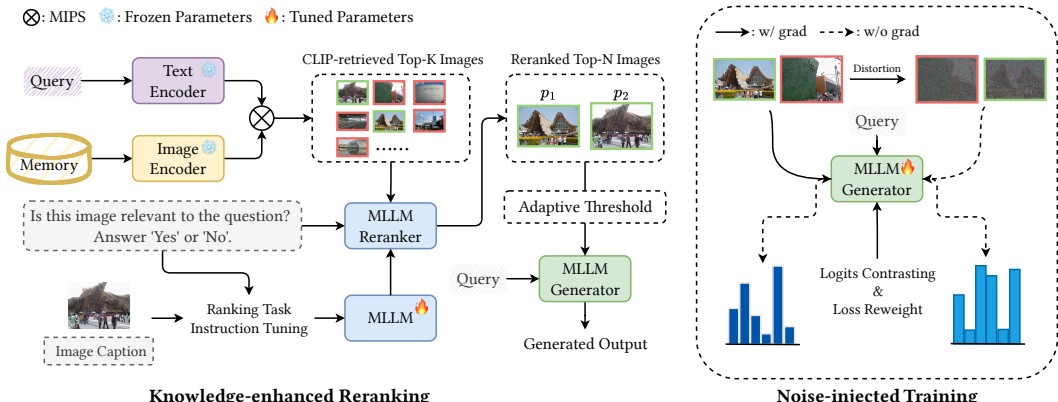

Figure 2: Overview of our proposed RagVL. In the retrieval stage, we utilize the CLIP model and faiss to find the top-$K$ most relevant images through Maximum Inner Product Search (MIPS) (Guo et al., 2020). Subsequently, the highly similar top-$K$ images are reranked into top-$N$ with the fine-tuned MLLM reranker. Finally, the top-$N$ images are fed into the MLLM generator along with the query for accurate generation.

encoder $\Phi_{img}$ to encode each image and build the image memory $\mathcal{M}$ using faiss (Douze et al., 2024). From the external memory $\mathcal{M}$, the retriever aims to retrieve a small set of images that support the textual query $q$. Specifically, we encode the query with the text encoder $\Phi_{text}$ and use MIPS over all of the image candidates $v \in \mathcal{M}$ as follows,

$$\hat{\mathcal{M}} = TopK(\mathcal{M}|q) = \underset{v \in \mathcal{M}}{TopK} \ \Phi_{text}(q) \cdot \Phi_{img}(v). \tag{1}$$

The top-$K$ images with the highest inner product scores, *i.e.* the nearest top-$K$ neighbors $\hat{\mathcal{M}} = \{v_1, v_2, \cdots, v_k\}$, are retrieved as the candidate images for answer generation.

### 3.3 INDUCING RANKING ABILITY OF MLLMS

CLIP stands out across a wide range of multimodal representations and retrieval tasks as a powerful and highly transferable model. However, when encountering long-tail distribution or domain-specific terms, CLIP fails to match the proper pairs across text and images. It results in lower accuracy and higher demand of $k$ value to increase the recall rate of supporting materials, which is time- and resource-consuming. More importantly, simply aligning different modalities is insufficient for real-time multimodal queries, leading to severe performance degradation in knowledge-intensive situations (see in Table 1). To mitigate this, we resort to MLLMs for their capabilities of semantic understanding. In general, MLLMs are pre-trained on vast image-text pairs for feature alignment and fine-tuned on language-image instruction-tuning datasets for instruction following. With this pre-injected multimodal knowledge, they are inherently capable of understanding semantically relevant contents across both visual and textual modalities at a deeper level, while CLIP merely computes similarity between vectors. Therefore, to mitigate the bottleneck challenge of multimodal RAG, we introduce the flexible knowledge-enhanced reranking to induce the ranking ability of MLLMs.

**Ranking Data Construction**  To enhance the ranking capability of MLLMs, we construct the instruction-following data based on each multimodal QA dataset. We treat each query and the ground truth images as relevant, while the hard negative images as irrelevant. We construct two types of ranking tasks and require the model to generate 'Yes' for the relevant pairs and 'No' for the irrelevant pairs. Intuitively, the caption-aware style brings more additional knowledge for the model to distinguish the relevance between the image and query. Therefore, we train the reranker with the caption-aware ranking task. In addition, the instruction tuning for ranking can be either blended into the supervised fine-tuning of downstream tasks or conducted separately. See the details of the instruction template in Table 9.

**Knowledge-enhanced Reranking**  By simply asking the question "*Based on the image and its caption, is the image relevant to the question? Answer 'Yes' or 'No'.*", we measure the relevance between the image and query with the probability $p$ of generating 'Yes' calculated from the output logits. Thus, reranking the top-$K$ candidates into top-$N$ can be formulated as follows,

$$\tilde{\mathcal{M}} = TopN(\hat{\mathcal{M}}|\phi) = \underset{(v,c) \in \hat{\mathcal{M}}}{TopN} \, p_\phi(v,c,q), \tag{2}$$

$$p_\phi(v,c,q) = \frac{\exp(logit(y_1 = \text{``Yes''}|v,c,q))}{\exp(logit(y_1 = \text{``Yes''}|v,c,q)) + \exp(logit(y_1 = \text{``No''}|v,c,q))}, \tag{3}$$

where $v$, $c$, and $q$ denote the image, corresponding caption, and query, respectively. $\phi$ is the weight of the reranker. $y_1$ denotes the first token in the generated output.

**Adaptive Threshold**  Since the reranked images might still have low relevance $p$ to the query, they can negatively affect answer generation, potentially performing worse than not including the images. To further improve the retrieval accuracy, we apply an adaptive threshold $\eta$ to filter out candidates when $p < \eta$. We set two types of thresholds: the natural threshold and the adaptive threshold. The natural threshold refers to $\eta = 0.5$, which is the natural boundary for our binary classification ranking. For more precise retrieval, we experiment on the validation set and utilize the intersection point of the interpolated curve of exact match and mismatch as the adaptive threshold. In this way, the model can avoid the distraction from irrelevant images. By forcing the MLLM to jointly consider the query, caption, and image, the simple yet effective question template stimulates and enhances the model's ranking ability with multimodal knowledge, thereby supporting the trustworthy generation.

### 3.4 NOISE-INJECTED TRAINING

Compared to providing a fixed number of images each time, the task with single/multiple images interleaved is more aligned with real-world scenarios. However, it also presents challenges in determining the optimal number of images to provide each time and in extracting relevant information rather than distracting information from the provided images. Though the reranker performs well in selecting relevant images, irrelevant ones still inevitably disturb the accurate generation.

Inspired by VCD (Leng et al., 2024), visual uncertainty amplifies language priors, and contrasting the logits obtained from the enhanced language priors with the original logits can better highlight visual relevance. In light of this, we propose enhancing the model's robustness by injecting visual noise during training, both at the data level and token level: (I) For single-image/multi-image interleaved datasets, we sample randomly from the hard negatives to ensure that each instruction-following data has the same amount of image input. (II) We introduce additional visual uncertainty by applying a Gaussian noise mask and contrasting the logits to reweight the loss for each token.

**Noise-injected Data Construction**  For datasets that may require both single and multiple image inputs, we standardize the number of image inputs for each sample in the instruction-following data to the maximum number needed for any question. In the case of WebQA, where each question requires 1-2 images for answering, we randomly sample 1 image from the hard negatives as an injected noise for the single-image query. The model is required to distinguish between relevant and irrelevant visual information, which strengthens its capability of visual understanding.

**Noise-injected Logits Contrasting**  Although injecting noise into the dataset can help the model better adapt to noisy environments, it can also be a double-edged sword, making the training process more unpredictable. Instead of the simple Maximum Likelihood Estimation (MLE) loss, we need a more robust objective (Xiao et al., 2024) to guide the model to learn the correlation between visual tokens and textual (query) tokens accurately. Thus, we resort to reweight the training loss and first employ the forward diffusion process (Ho et al., 2020) to distort the image:

$$f(v_t \mid v_{t-1}) = \mathcal{N}\left(v_t; \sqrt{1-\gamma}v_{t-1}, \gamma\mathbf{I}\right), \, f(v_T \mid v_0) = \prod_{t=1}^{T} f(v_t \mid v_{t-1}), \tag{4}$$

where $\mathbf{I}$ and $v_0$ denote an identity matrix and the original image, respectively. We gradually distort the original image by adding the Gaussian noise for $T$ steps and $\gamma$ controls the the amount of noise

added in each step. Subsequently, given a textual query $x$ and an image input $v$, the model generates two logit distributions conditioned on different visual posteriors: the original $v$ and distorted $v^*$. By contrasting the logit distributions obtained from these two conditions, we can get the contrastive probability distribution of the $i$-th sample at time step $t$ as follows,

$$\Delta logit(y_{i,t}|v_i, v_i^*, x_i, y_{i,<t}) = logit_\theta(y_{i,t}|v_i, x_i, y_{i,<t}) - logit_\theta(y_{i,t}|v_i^*, x_i, y_{i,<t}), \quad (5)$$

where $y_{i,t}$ and $y_{i,<t}$ denote the token at time step $t$ and the generated tokens sequence up to the time step $t-1$ of the $i$-th sample, respectively. Subsequently, we can obtain the visual correlation weight:

$$\mathbf{w}_{i,t} = \Delta logit(y_{i,t}|v_i, v_i^*, x_i, y_{i,<t}). \quad (6)$$

Following Xiao et al. (2024) to post-process and smooth the weights, we finally reassign the weight of each token in the vanilla MLE loss, which can be formulated as follows,

$$\mathcal{L}_{INJ}^{i,t} = -\frac{\tilde{\mathbf{w}}_{i,t}}{\sum_{k=1}^{l} \tilde{\mathbf{w}}_{i,k}} \cdot logp_\theta(y_{i,t}|v_i, x_i, y_{i,<t}), \quad (7)$$

where $l$ and $\tilde{\mathbf{w}}$ represent the length of textual tokens and the smooth weight, respectively.

## 4 EXPERIMENTS AND ANALYSIS

### 4.1 EXPERIMENT SETUP

**Datasets and Evaluation Metrics**  For evaluation, we consider the image-related subsets of two multimodal QA datasets WebQA and MultimodalQA. Both datasets contain multimodal knowledge-seeking query-answer pairs. Since the test set labels from both datasets are not publicly available, the training and validation sets in our work are new subsets of the original training data, while the test sets are sourced from the original validation sets. Each query is associated with a set of hard negative distractors so that two evaluation setups can be used, namely distractor and full-wiki. However, we only consider the full-wiki setting to demonstrate the superiority of our *retrieval-rerank-generation* pipeline. Additionally, we conduct more experiments on Flickr30K (Young et al., 2014) and MS-COCO (Lin et al., 2014) to evaluate the performance on caption-to-image retrieval tasks. More details can be found in Appendix A, B and E.

### 4.2 EVALUATION ON MULTIMODAL KNOWLEDGE-SEEKING

**Results of Retrieval**  Table 1 shows the performance of knowledge-enhanced rerankers on Mulit-modalQA and WebQA. The experimental results show that the retriever performs weakly regarding precise recall (R@1 and R@2) on both datasets, making it difficult for accurate multimodal retrieval-augmented generation. Since the captions from the two datasets are basically names of objects or places, it isn't easy to adapt to the scenarios using vanilla contrastive learning, as proven in the table. After inducing the ranking abilities of MLLMs, our proposed method effectively improves the retrieval performance by a large margin. Specifically, with five MLLMs, our method consistently improves R@2 on WebQA by an average of 40%. The results are significantly improved after reranking the Top-K candidates from four different retrievers. Notably, on MultimodalQA, it reaches the upper limit of Recall@20 (98.26%) from CLIP on LLaVA-v1.5-13B and InternVL2-1/2B. These demonstrate the superior performance of our proposed method in multimodal knowledge retrieval.

**Generalizabilities of Caption-aware Instruction Tuning**  To further validate the generalizability of our method, on one hand, we test the reranker fine-tuned on the WebQA training set on MMQA. As shown in Figure 3a, the reranker trained on WebQA exhibits competitive performance in terms of R@1, R@5, and R@10, and even matches the original reranker's performance with InternVL2-1/2B. On the other hand, we randomly select different portions of data from WebQA to train InternVL2-2B in a low-resource setting, and obtain the probability distribution of the reranker outputing '*Yes*' for correctly recalled images. Figure 3b shows the robust performance of our proposed method under the low-resource settings. With only 2.5% of the original data, the reranker significantly outperforms the strong retriever baseline, InternVL-G, in terms of R@2. As the data scale increases, the probability of correctly recalling images also improves, stabilizing around 20%, and the recall metrics follow a similar trend. In summary, these two points fully demonstrate the strong generalizability of our proposed method across datasets as well as in low-resource settings within the same dataset, making it easily adaptable to more scenarios. We provide an evaluation on LLaVA-v1.5-13B and make further discussion in Appendix F.

Table 1: Performance of knowledge-enhanced rerankers on multimodal knowledge-seeking. The reranking is conducted based on the top 20 candidates from the retrievers (see details in Appendix A). The best scores in each setting are in **bold**.

| Methods | MultimodalQA | | | WebQA | | |
|---|---|---|---|---|---|---|
| | R@1 | R@5 | R@10 | R@2 | R@5 | R@10 |
| CLIP-ViT-L/14-336px | **84.78** | 94.35 | 95.65 | 57.10 | 71.96 | 84.86 |
| *w/ SFT* | 83.04 | 94.35 | 94.78 | 55.09 | 73.23 | 81.94 |
| Vis-BGE-base | 49.57 | 74.78 | 82.61 | 28.78 | 43.62 | 54.56 |
| Vis-BGE-m3 | 43.48 | 66.52 | 72.17 | 26.69 | 40.75 | 51.14 |
| InternVL-C | 82.17 | **95.65** | 96.96 | **64.90** | **81.22** | 88.09 |
| InternVL-G | 82.17 | 95.22 | **97.39** | **64.90** | 80.23 | **88.28** |
| *Reranking Top-K from CLIP-ViT-L/14-336px* | | | | | | |
| LLaVA-v1.5-13B | 72.61 | 90.87 | 95.22 | 45.35 | 65.87 | 80.56 |
| *w/ caption-aware* IT | **98.26** | **98.26** | **98.26** | 79.74 | 88.14 | 89.77 |
| mPLUG-Owl2 | 67.83 | 87.39 | 93.91 | 43.26 | 63.80 | 79.38 |
| *w/ caption-aware* IT | 90.87 | 96.09 | 97.39 | 71.27 | 85.08 | 88.97 |
| Qwen-VL-Chat | 68.26 | 89.57 | 92.61 | 47.64 | 67.22 | 80.42 |
| *w/ caption-aware* IT | 91.30 | 95.65 | 97.39 | 80.12 | 88.53 | **89.96** |
| InternVL2-1B | 47.39 | 84.78 | 93.91 | 34.99 | 57.49 | 74.72 |
| *w/ caption-aware* IT | **98.26** | **98.26** | **98.26** | **82.00** | 88.78 | 89.94 |
| InternVL2-2B | 66.52 | 88.70 | 93.91 | 42.79 | 62.48 | 77.97 |
| *w/ caption-aware* IT | **98.26** | **98.26** | **98.26** | 81.91 | **88.94** | 89.94 |
| *Reranking Top-K from Different Retrievers* | | | | | | |
| LLaVA-v1.5-13B | | | | | | |
| *w/ Vis-BGE-base* | 88.70 | 88.70 | 88.70 | 59.61 | 64.71 | 65.70 |
| *w/ Vis-BGE-m3* | 84.78 | 84.78 | 84.78 | 57.57 | 62.26 | 63.03 |
| *w/ InternVL-C* | **98.70** | **98.70** | **98.70** | **82.08** | **90.79** | **92.72** |
| *w/ InternVL-G* | 97.83 | 97.83 | 97.83 | 81.91 | 90.24 | 92.31 |

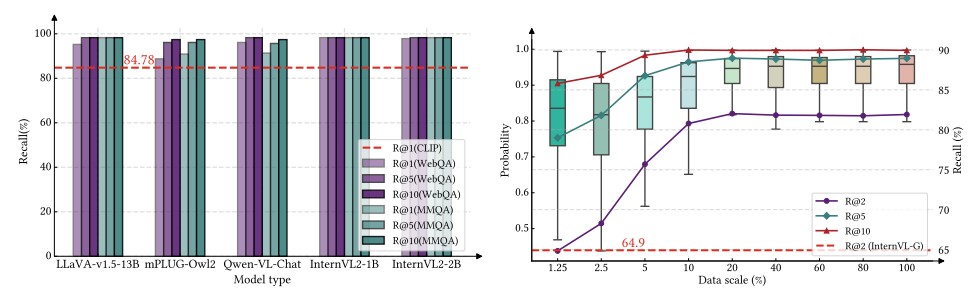

(a) Generalizability of reranking.    (b) Low-resource settings on WebQA.

Figure 3: Generalizabilities of caption-aware instruction tuning. (a) compares the performance of the reranker fine-tuned on WebQA with the one fine-tuned on MultimodalQA, evaluated on MultimodalQA. (b) visualizes the changes in the probability distribution of correctly recalled items and the recall rate of the reranker under low-resource settings as the scale of the training dataset varies.

## 4.3 EVALUATION ON MULTIMODAL RETRIEVAL-AUGMENTED GENERATION

**Reranking Performance with Thresholds** Due to the strong performance of the reranker in low-resource settings, we train InternVL2-1/2B as the rerankers using only 20% of the data, considering efficiency. As shown in Figure 4, we collect the relevance probability of the image candidates after reranking and the results prove the superiority of our proposed knowledge-enhanced reranking. Among the train, validation, and test sets, the relevance probabilities of correct recalls are concentrated in the highest range. Since there is still a portion of erroneous recalls, we plotted the interpo-

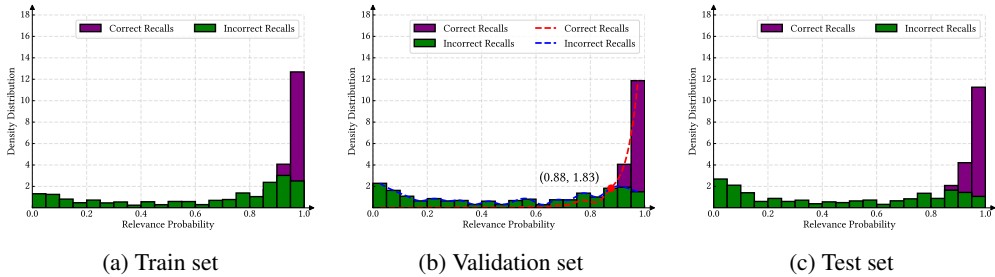

(a) Train set          (b) Validation set          (c) Test set

Figure 4: Density distribution of the relevance probability of correct and incorrect recalls on WebQA after reranking from the InternVL2-2B reranker.

Table 2: Performance of InternVL2-2B reranker on two benchmark datasets. *P* and *R* denote precision and recall, respectively. The best scores in each setting are in **bold**.

| Methods | MultimodalQA | | | WebQA | | |
|---|---|---|---|---|---|---|
| | P | R | F1 | P | R | F1 |
| CLIP Top-$N$ | 84.78 | 84.78 | 84.78 | 41.24 | 57.10 | 47.89 |
| *Caption-aware Instruction Tuning* | | | | | | |
| CLIP Top-$K$ + Reranker | 98.26 | **98.26** | 98.26 | 59.26 | **82.05** | 68.82 |
| *w/* Natural Threshold | **100.00** | 97.83 | **98.90** | 74.89 | 80.59 | **77.64** |
| *w/* Adaptive Threshold | **100.00** | 97.83 | **98.90** | **88.34** | 68.29 | 77.03 |

lated curves of correct recalls and erroneous recalls on the validation set and took the x-coordinate of their intersection point as the adaptive threshold. Due to the perfect performance on MultimodalQA with the natural threshold, we set the adaptive threshold to the same as the natural threshold.

As demonstrated in Table 2, our proposed knowledge-enhanced reranking method demonstrates superior performances. We train the reranker with the caption-aware instructions and achieve better performance across all metrics compared to directly using CLIP for top-$N$ retrieval. When the adaptive threshold $\eta$ is activated, the model accurately filters out irrelevant images, improving *accuracy* and *F1* score. Specifically, in WebQA, when $\eta$ is set to an intuitively reasonable value of 0.5, the corresponding F1 score increases by 29.75%. In MultimodalQA, the reranker successfully identifies all ground truth images from the retrieved top-$K$ candidates when $\eta$ is set to 0.5, proving the strong capability of our proposed method in retrieval reranking.

**Results of Retrieval-augmented Generation**   Table 3 displays the results on multimodal question answering which requires retrieving images. The baselines without retrieval show limited performance, even the powerful gpt-3.5-turbo-0125 fails to answer the knowledge-intensive questions. Notably, the backbone LLMs of InternVL2-1/2B (Qwen2-0.5B-Instruct and internlm2-chat-1_8b) perform poorly while their multimodal counterparts are somewhat improved. This phenomenon indicates that MLLMs can indeed learn world knowledge from different modalities and store it inside the parameters. Moreover, as a more timely and flexible approach for knowledge updates, retrieval-augmented generation also offers the potential for better knowledge integration in MLLMs.

After applying our proposed pipeline, all configurations on InternVL2-1B and InternVL2-2B demonstrate excellent performance of retrieval-augmented generation, approaching or even surpassing the performance of *Oracle*. When the natural threshold is activated, there is a significant increase in the accuracy of recalling the correct images (as shown in Table 2), leading to substantial improvements in all metrics across both datasets. Moreover, this improvement is more evident in the single-image scenario. This is because we fixed the number of images recalled each time, and setting the threshold allows us to filter out erroneously recalled images, resulting in a consistent performance enhancement. However, when adopting adaptive thresholds, the improvement in results is not as significant as with natural thresholds. This can be seen from Table 2, where, despite a substantial

Table 3: Performance of multimodal knowledge-seeking question answering on WebQA and Multi-modalQA. In addition to the overall results, we report the accuracy of single-image and multi-image input with *Single.* and *Multi.* for WebQA, respectively. *Oracle* refers to directly feeding the ground truth image to the generator after *NIT (Noise-injected Training)*. The best scores in each setting are in **bold**.

| Methods | MultimodalQA | WebQA | | |
|---|---|---|---|---|
| | EM | Single. | Multi. | Overall |
| *w/o Retrieval-augmented Generation* | | | | |
| Qwen2-0.5B-Instruct | 10.43 | 17.29 | 19.33 | 18.20 |
| internlm2-chat-1_8b | 10.43 | 23.25 | 32.58 | 27.40 |
| gpt-3.5-turbo-0125 | **25.22** | **40.80** | **54.49** | **46.88** |
| InternVL2-1B | 19.57 | 26.10 | 43.57 | 33.86 |
| InternVL2-2B | **25.22** | 30.37 | 48.20 | 38.29 |
| *InternVL2-1B w/ Retrieval-augmented Generation* | | | | |
| InternVL2-1B | | | | |
|   *w/ CLIP Top-N* | 50.87 | 35.98 | 48.65 | 41.61 |
|   *w/ InternVL-G Top-N* | 49.57 | 38.88 | 49.11 | 43.43 |
| **RagVL** *w/o NIT* | 54.78 | 38.09 | 50.91 | 43.79 |
|   *w/ Natural Threshold* | 54.78 | 40.43 | 50.96 | 45.11 |
|   *w/ Adaptive Threshold* | 54.78 | 40.64 | 50.98 | 45.23 |
| **RagVL** *w/ NIT* | 68.26 | 53.07 | 72.53 | 61.72 |
|   *w/ Natural Threshold* | **68.70** | 56.68 | 72.49 | 63.71 |
|   *w/ Adaptive Threshold* | **68.70** | **56.71** | **72.60** | **63.78** |
| Oracle | 69.13 | 60.09 | 73.23 | 65.93 |
| *InternVL2-2B w/ Retrieval-augmented Generation* | | | | |
| InternVL2-2B | | | | |
|   *w/ CLIP Top-N* | 61.30 | 40.80 | 48.88 | 44.39 |
|   *w/ InternVL-G Top-N* | 60.00 | 41.92 | 48.45 | 44.82 |
| **RagVL** *w/o NIT* | 64.78 | 41.68 | 48.40 | 44.67 |
|   *w/ Natural Threshold* | 65.65 | 44.71 | 48.97 | 46.60 |
|   *w/ Adaptive Threshold* | 65.65 | 44.37 | 48.98 | 46.42 |
| **RagVL** *w/ NIT* | 73.04 | 53.91 | 72.62 | 62.23 |
|   *w/ Natural Threshold* | 73.48 | 57.25 | **73.01** | 64.25 |
|   *w/ Adaptive Threshold* | **73.48** | **57.94** | 72.47 | **64.40** |
| Oracle | 73.48 | 60.66 | 73.59 | 66.41 |

increase in accuracy, there is a significant drop in recall. Therefore, natural thresholds are a better and more efficient choice for retrieval-augmented generation.

**Ablation Studies** To validate the efficacy of each component in our proposed method, we conduct a set of ablation experiments on We-bQA with InternVL2-2B, and the results are reported in Table 4. For "*w/o Reranker*", we directly retrieve Top-2 images with CLIP in the inference stage. The use of the reranker in RagVL shows an improvement in all metrics (*Single.*, *Multi.* and *Overall*) compared to "*w/o Reranker*". For "*w/o ND*", we replace the noise-injected dataset with the vanilla dataset. Ablating *ND* results in a performance decrease on all metrics. Introducing noise at both data and token levels helps the model learn to distin-

Table 4: Ablation study on WebQA with InternVL2-2B. *NLC* and *ND* refer to Noise-injected Logits Contrasting and Noise-injected Data, respectively.

| Methods | WebQA | | |
|---|---|---|---|
| | Single. | Multi. | Overall |
| **RagVL** ($\eta = 0.5$) | **57.25** | **73.01** | **64.25** |
|   *w/o* Reranker | 53.63 | 71.79 | 61.70 |
|   *w/o* ND | 57.11 | 71.24 | 63.39 |
|   *w/o* NLC | 56.42 | 72.40 | 63.52 |
|   *w/o* ND & NLC | 56.27 | 70.10 | 62.42 |

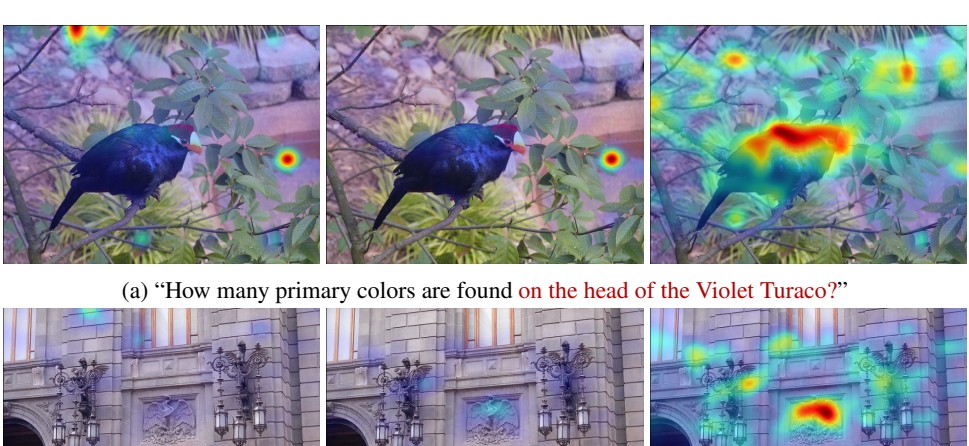

(a) "How many primary colors are found on the head of the Violet Turaco?"

(b) "Which is better maintained, the carving on the front of the Palace of the Governor in Uxmal or the Bird carving above the doorway in Mexico, Architecture?"

Figure 5: Visualization of attention heatmaps *w/* and *w/o NIT*. Displayed from left to right are the attention maps for the base model (*w/o IT*), the model fine-tuned *w/o NIT*, and the model fine-tuned *w/ NIT*, respectively, with each corresponding to its respective question in the caption.

guish between the candidate images more effectively in multi-image inference. Since *NLC* enhances the model's robustness at the token level, ablating it leads to a decrease in all metrics on WebQA. This decline is more pronounced when both *NLC* and *ND* are ablated, especially in multi-image inference scenarios. Therefore, our proposed training method, which injects noise at the data and token levels, helps reduce the distractions from noise and mitigate MNC during inference time.

## 4.4 QUALITITIVE ANALYSIS

As illustrated in Figure 5, we visualize the attention heatmaps from the three models, providing insights into how each model focuses on the details of the input image. The attention weights are calculated by accumulating the attention score between image tokens and text tokens across all layers. Obviously, the model *w/ NIT* provides more focused attention on the crucial parts of the query than the other two models. More cases can be seen in Appendix I.

## 5 CONCLUSION

In this paper, we present a robust framework for enhancing Multimodal Large Language Models (MLLMs) through knowledge-enhanced reranking and noise-injected training to tackle the multi-granularity noisy correspondence (MNC) problem in multimodal retrieval-augmented generation. Our comprehensive approach addresses both coarse-grained and fine-grained noise, significantly improving retrieval accuracy and generation robustness. The results from our extensive experiments on the WebQA and MultimodalQA datasets demonstrate the superiority of our proposed method, especially in scenarios requiring fine-grained visual understanding and robust generation. By instruction-tuning MLLMs for reranking and injecting visual noise during training, we enhance the model's capability to handle real-world noisy data and improve its overall performance in multimodal tasks.

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

## A  DATA STATISTICS AND EVALUATION METRICS

Table 5: Overall statistics of datasets.

| Dataset | Train | Dev | Test |
|---|---|---|---|
| WebQA | 15K | 3.7K | 2.5K |
| MultimodalQA | 2K | 420 | 230 |
| Flickr30K | 29K | 1K | 1K |
| MS-COCO | 113K | 5K | 5K |

Table 6: Recall@20 of different retrievers.

| Methods | MultimodalQA | WebQA | Flickr30K | MS-COCO |
|---|---|---|---|---|
| CLIP-ViT-L/14-336px | 98.26 | 90.27 | 96.54 | 96.84 |
| Vis-BGE-base | 88.70 | 65.89 | 93.64 | 95.86 |
| Vis-BGE-m3 | 84.78 | 63.14 | 91.48 | 91.98 |
| InternVL-C | 98.70 | 93.27 | 98.92 | 98.64 |
| InternVL-G | 97.83 | 92.78 | 99.22 | 99.02 |

**WebQA** consists of queries requiring 1-2 images or text snippets, while 44% of image-based and 99% of text-based queries need multiple knowledge sources. Following the vanilla evaluation setting, we measure the overlap of key entities between the generated output and ground truth answer as *Accuracy*.

**MultimodalQA** contains multimodal questions over tables, text, and images. We focus on the QA pairs requiring only image information, which are annotated as 'ImageQ' and attached to 1 image each. The evaluation metric used is Exact Match (*EM*).

**Flickr30K** consists of 31,000 images sourced from Flickr, each accompanied by five captions. Consistent with the setup of Lee et al. (2018), we allocate 1,000 images for validation, 1,000 for testing, and use the remaining images for training.

**MS-COCO** comprises 123,287 images, each paired with five captions. Following the protocol in Lee et al. (2018), we designate 113,287 images for training, 5,000 for validation, and 5,000 for testing.

## B    IMPLEMENTATION DETAILS

To evaluate the effectiveness and generalizability of our proposed method, this paper leverages several cutting-edge MLLMs as the backbone, including LLaVA-v1.5-13B (Liu et al., 2024a), mPLUG-Owl2 (Ye et al., 2024), Qwen-VL-Chat (Bai et al., 2023), and InternVL (Chen et al., 2024). We employ the frozen CLIP-ViT-L/14-336px as the vision and text encoder. For *RagVL*, we first train the reranker model with the caption-aware ranking task. Subsequently, we use CLIP to retrieve top-$K$ candidates and rerank them into top-$N$ with the fine-tuned reranker. $K$ is set to 20, while $N$ is set to 2 for WebQA and 1 for MultimodalQA. All trainings are conducted under the LoRA (Hu et al., 2021) setting. For evaluation, we use greedy decoding to ensure reproducibility and report the best performance. All experiments are conducted on 8 40G NVIDIA A100 GPUs.

## C    COMPUTATIONAL EFFICIENCY

Table 7 presents the inference time for different settings on 4 A100 GPUs. As shown, *"CLIP Top-K"* only requires a small amount of time due to fast inner product search, while our proposed method requires more time on reranking the retrieved candidates. Though the MLLM reranker shows powerful retrieval performance, the efficiency will be a major issue that limits its development.

Thanks to advances in inference acceleration, we can address the efficiency issue from different perspectives. For example, FlashAttention (Dao et al., 2022) enables faster inference with lower resources by using tiling to reduce the number of memory reads/writes between GPU memories. PagedAttention (Kwon et al., 2023) resorts to the classical virtual memory and paging techniques in operating systems to achieve near-zero waste and flexible sharing of KV cache memory. To be more specific, we can share the attention calculation of textual tokens

Table 7: Inference time per sample. Each inference with the reranker involves 20 evaluations of image relevance and one generation of an answer.

| Approach | Time Cost |
|---|---|
| CLIP Top-K | 1.23s |
| + InternVL2-2B reranker | 5.11s |
| + LLaVA-v1.5-13B reranker | 6.24s |

among different candidates and parallelize the computation of visual tokens to maximize resource utilization and accelerate inference since the textual instructions of all candidates during the reranking process are identical. As a successful attempt, Prompt Cache (Gim et al., 2024) has made similar efforts to reduce latency in time-to-first-token, which improves 8x for GPU-based inference and maintains output accuracy.

## D    EFFECT OF CAPTIONS

We conduct experiments on test sets of WebQA ranking and QA datasets to verify the validity of captions in retrieving relevant sources. In WebQA QA task, we retrieve top-20 candidate images using CLIP and rerank them into top-2 with our instruction-tuned reranker models. As shown in Table 8, the vanilla LLaVA-v1.5-13B performs poorly on both tasks. The models trained on the ranking task outperform the baseline, particularly the one trained on the caption-aware task. This demonstrates the superiority of our simple yet effective instruction templates in inducing the ranking ability of MLLMs.

Table 8: Reranking performance of different models on WebQA.

| Methods | WebQA Ranking | WebQA QA |
|---|---|---|
| | Acc | Recall@2 |
| CLIP-ViT-L/14-336px | - | 57.10 |
| LLaVA-v1.5-13B | 67.74 | 45.35 |
| *w/ caption-agnostic* IT | 89.62 | 54.45 |
| *w/ caption-aware* IT | **93.99** | **79.74** |

## E    PERFORMANCE ON CAPTION-TO-IMAGE RETRIEVAL

To further verify the effectiveness and generalizability of our proposed reranking method, we conduct more experiments on Flickr30K and MS-COCO. We construct the reranking tasks and prompt

Table 9: The instruction template for ranking and generation tasks. The retrieval-augmented QA task allows multi-image input, whereas the ranking tasks consider one image at a time.

| Task | Instruction | Answer |
|---|---|---|
| Multimodal Retrieval-augmented QA | <image> ··· <image> {question} | A phrase |
| Caption-agnostic Ranking | <image> Question:{question} Is this image relevant to the question? Answer 'Yes' or 'No'. | Yes / No |
| Caption-aware Ranking (QA) | <image> Image Caption:{caption} Question:{question} Based on the image and its caption, is the image relevant to the question? Answer "Yes" or "No". | Yes / No |

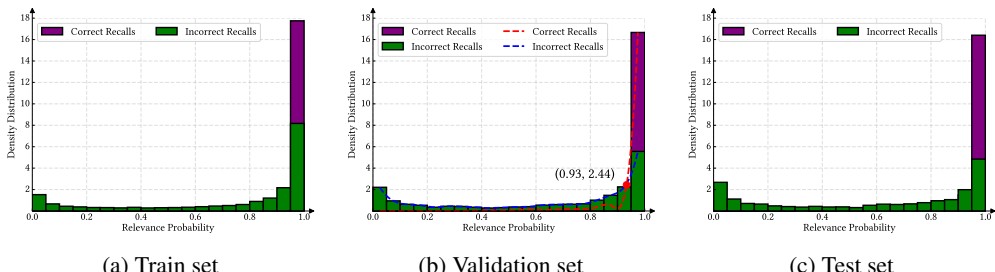

(a) Train set    (b) Validation set    (c) Test set

Figure 7: Density distribution of the relevance probability of correct and incorrect recalls on WebQA after reranking from the LLaVA-v1.5-13B reranker.

the reranker with the instruction "*<image> Image Caption:{caption} Is the image relevant to the caption? Answer 'Yes' or 'No'*". As shown in Table 10, our proposed method still outperforms the majority of existing retrievers across all metrics, except for InternVL-G, which is specifically designed for image-text matching. Our approach primarily focuses on cases where the query is a question, and the keys are captions and images. In contrast, in these two caption-to-image retrieval datasets, the query is a caption, and the key is an image. Thus, our method not only demonstrates superior performance in multimodal RAG but also maintains generalizability and competitiveness in traditional text-to-image retrieval.

# F   MORE EVALUATIONS ON LLAVA-V1.5-13B

**Low-resource Settings on WebQA** As shown in Figure 6, the experiments with LLaVA-v1.5-13B under low-resource settings also verified the robustness of our proposed method in reranker training. With only 2.5% of the original data, the reranker significantly surpasses the original baseline, InternVL-G, in R@2 and almost reaches the performance peak. This inspires us to further explore the performance of low-resource instruction fine-tuning for models with different parameter sizes in future work, aiming to enhance the generalizability and efficiency of MLLMs in instruction fine-tuning and downstream task deployment.

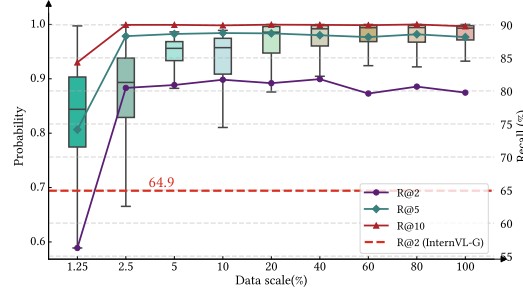

Figure 6: Retrieval performance on WebQA with LLaVA-v1.5-13B under low-resource settings.

**Reranking Performance with Thresholds**
Similarly, we train LLaVA-v1.5-13B as the reranker using only 20% of the data. As shown in Figure 7, the relevance probabilities of correct recalls are concentrated in the highest range. The adaptive threshold is high enough to filter out most of the incorrect candidates.

Table 10: Performance of knowledge-enhanced rerankers on caption-to-image retrieval. The best scores in each setting are in **bold**.

| Methods | Flickr30K | | | MS-COCO | | |
|---|---|---|---|---|---|---|
| | R@1 | R@5 | R@10 | R@1 | R@5 | R@10 |
| CLIP-ViT-L/14-336px | 66.90 | 89.00 | 93.36 | 57.18 | 83.24 | 91.90 |
| Vis-BGE-base | 57.38 | 83.28 | 89.60 | 52.94 | 81.22 | 90.12 |
| Vis-BGE-m3 | 52.18 | 78.18 | 86.06 | 43.14 | 73.44 | 84.42 |
| InternVL-C | 81.50 | 95.94 | 97.82 | 71.82 | 92.06 | 96.62 |
| InternVL-G | **84.28** | **96.88** | **98.44** | **76.20** | **94.24** | **97.54** |
| *Reranking Top-K from CLIP-ViT-L/14-336px* | | | | | | |
| LLaVA-v1.5-13B | 79.90 | 94.52 | 96.24 | 71.10 | 92.02 | 95.96 |
| *w/ caption-aware* IT | 83.04 | **95.34** | 96.34 | 74.64 | 93.16 | 95.62 |
| mPLUG-Owl2 | 76.16 | 94.12 | 95.98 | 65.44 | 90.34 | 95.38 |
| *w/ caption-aware* IT | 81.38 | 94.70 | 96.08 | 69.96 | 91.30 | 95.36 |
| Qwen-VL-Chat | 82.70 | 94.80 | 96.26 | 74.40 | 92.72 | 95.98 |
| *w/ caption-aware* IT | **84.40** | 95.18 | 96.30 | **76.62** | **93.56** | **96.26** |
| InternVL2-1B | 67.74 | 92.56 | 96.04 | 55.76 | 87.14 | 94.02 |
| *w/ caption-aware* IT | 83.02 | 95.12 | **96.38** | 74.24 | 92.78 | 96.02 |
| InternVL2-2B | 67.74 | 92.56 | 96.04 | 71.32 | 92.06 | 95.82 |
| *w/ caption-aware* IT | 83.78 | 95.14 | 96.32 | 75.86 | 93.40 | 96.10 |
| *Reranking Top-K from Different Retrievers* | | | | | | |
| LLaVA-v1.5-13B | | | | | | |
| *w/ Vis-BGE-base* | 80.76 | 92.56 | 93.44 | 74.12 | 92.36 | 95.02 |
| *w/ Vis-BGE-m3* | 79.64 | 90.46 | 91.34 | 71.94 | 88.96 | 91.18 |
| *w/ InternVL-C* | **83.56** | 97.12 | 98.58 | 75.00 | 94.26 | 97.36 |
| *w/ InternVL-G* | 83.26 | **97.16** | **98.80** | **75.06** | **94.36** | **97.60** |

Table 11: Performance of LLaVA-v1.5-13B reranker on two benchmark datasets. *P* and *R* denote precision and recall, respectively. The best scores in each setting are in **bold**.

| Methods | MultimodalQA | | | WebQA | | |
|---|---|---|---|---|---|---|
| | P | R | F1 | P | R | F1 |
| CLIP Top-$N$ | 84.78 | 84.78 | 84.78 | 41.24 | 57.10 | 47.89 |
| *Blended Instruction Tuning* | | | | | | |
| CLIP Top-$K$ + Reranker | 98.26 | **98.26** | 98.26 | 57.05 | **78.99** | 66.25 |
| *w/ Natural Threshold* | **100.00** | 97.39 | **98.68** | 67.94 | 78.00 | **72.62** |
| *w/ Adaptive Threshold* | **100.00** | 97.39 | **98.68** | **84.13** | 62.70 | 71.85 |
| *Ranking-only Instruction Tuning* | | | | | | |
| CLIP Top-$K$ + Reranker | 98.26 | **98.26** | 98.26 | 57.59 | **79.74** | 66.87 |
| *w/ Natural Threshold* | **100.00** | 97.83 | **98.90** | 68.31 | 78.52 | 73.06 |
| *w/ Adaptive Threshold* | **100.00** | 97.83 | **98.90** | 80.38 | 68.35 | **73.88** |

As shown in Table 11, our proposed knowledge-enhanced reranking method demonstrates superior performances. We train the reranker under two settings: (i)Blended training of ranking and QA tasks. (ii) Training exclusively with the ranking task. Whether training with the blended or separate setting, our approach achieves better performance across all metrics than directly using CLIP for top-$N$ retrieval. When the adaptive threshold $\eta$ is activated, the model accurately filters out irrelevant images, resulting in improved *accuracy* and *F1* score. Specifically, in WebQA, when $\eta$ is set to an intuitively reasonable value of 0.5, the corresponding F1 score increases by 25.17% after training on the ranking-only task. In MultimodalQA, the reranker successfully identifies all ground truth

Table 13: Performance of multimodal question answering on two benchmark datasets requiring image retrieval. In addition to the overall results, we report the accuracy of single-image and multi-image input with *Single.* and *Multi.* for WebQA, respectively. *Oracle* refers to directly feeding the ground truth image to the generator. The best scores in each training setting are in **bold**.

| Methods | MultimodalQA | WebQA | | |
|---|---|---|---|---|
| | EM | Single. | Multi. | Overall |
| *w/o Retrieval-augmented Generation* | | | | |
| Vicuna-v1.5-13B | 8.26 | 32.43 | 42.82 | 37.05 |
| Llama-2-13b-chat-hf | 0.43 | 16.23 | 21.27 | 18.47 |
| LLaVA-v1.5-13B | 42.61 | 31.92 | 50.37 | 40.12 |
| *LLaVA-v1.5-13B w/ Retrieval-augmented Generation* | | | | |
| LLaVA-v1.5-13B | | | | |
|   *w/* CLIP Top-$N$ | 75.65 | 41.29 | 47.54 | 44.07 |
|   *w/* InternVL-G Top-$N$ | 75.22 | 42.37 | 47.71 | 44.74 |
| **RagVL** *w/o NIT* | 78.70 | 41.03 | 48.09 | 44.17 |
|   *w/* Natural Threshold | 79.57 | 44.50 | 48.47 | 46.26 |
|   *w/* Adaptive Threshold | 79.57 | 44.05 | 49.00 | 46.25 |
| **RagVL** *w/ NIT* | 78.70 | 57.06 | 76.18 | 65.56 |
|   *w/* Natural Threshold | **79.57** | 60.86 | 76.83 | 67.95 |
|   *w/* Adaptive Threshold | **79.57** | **61.76** | **76.90** | **68.49** |
| Oracle | 79.13 | 65.51 | 77.04 | 70.63 |

images from the retrieved top-$K$ candidates when $\eta$ is set to 0.5, proving the strong capability of our proposed method in retrieval reranking.

For "*w/ Blended Reranker*", we utilize the blended reranker for both reranking and generation, which is trained with noise-injected data and vanilla MLE loss. Though we directly mix the ranking and QA datasets due to a lack of sufficient datasets, the blended reranker still performs competitively. Since training the blended reranker requires precise adjustments (Yu et al., 2024) to the composition of the training datasets to achieve better results, the results show a promising direction for future research (unifying reranker and generator), which further demonstrates the generalizability and superiority of our proposed method.

**Results of Retrieval-augmented Generation**
Table 13 displays the results of LLaVA-v1.5-13B on MultimodalQA and WebQA. Our proposed approach still outperforms baselines on all configurations. Due to a larger amount of parameters, LLaVA-v1.5-13B outperforms InternVL2-1/2B in answer generation. What's more, the adaptive threshold works better on LLaVA-v1.5-13B because the relevance probabilities of correct recalls are more focused in the high range. Therefore, our proposed method is also applicable to models with larger parameters.

Table 12: Ablation study on WebQA with LLaVA-v1.5-13B. *NLC* and *ND* refer to Noise-injected Logits Contrasting and Noise-injected Data, respectively.

| Methods | WebQA | | |
|---|---|---|---|
| | Single. | Multi. | Overall |
| **RagVL** ($\eta = 0.5$) | 60.86 | **76.83** | **67.95** |
|   *w/o* Reranker | 58.67 | 75.66 | 66.22 |
|   *w/o* ND | **61.67** | 75.19 | 67.68 |
|   *w/o* NLC | 60.08 | 76.24 | 67.26 |
|   *w/o* ND & NLC | 60.68 | 74.92 | 67.01 |
|   *w/* Blended Reranker | 58.15 | 74.97 | 65.63 |

**Ablation Studies** As shown in Table 12, we ablate the proposed approaches on WebQA with LLaVA-v1.5-13B. Similar to the results from InternVL2-2B, the benefits from reranking and noise injection are still significant. Specially, to explore the possibility of unifying reranker and generator, we utilize the blended reranker for both retrieval and generation. The results are very promising, and there is still significant room for optimization.

## G    EVALUATION ON GENERAL BENCHMARK DATASETS

While training a model on specific tasks can reduce its generalization capabilities (Ling et al., 2023), a moderate trade-off in universality is often acceptable to significantly enhance task-specific performance. As demonstrated in Table 14, we evaluated our approach on three general datasets: MME (Fu et al., 2024), MMBench (Liu et al., 2025), and SEED-Image (Li et al., 2024). Following noise-injected fine-tuning on WebQA, performance declined only marginally—by 5.2%–5.5%, 1.6%–2.5%, and 1.4%–1.9% on MME, MMBench, and SEED-Image, respectively. However, this fine-tuning

Table 14: Evaluation on three general benchmark datasets.

| Models | MME | MMBench EN-test | SEED Image |
|---|---|---|---|
| InternVL2-1B | 1769.2 | 61.72 | 65.60 |
| *w/* WebQA NIT | 1671.3 | 60.76 | 64.32 |
| InternVL2-2B | 1839.8 | 72.25 | 71.60 |
| *w/* WebQA NIT | 1743.2 | 70.46 | 70.60 |

resulted in a substantial improvement of approximately 40% on WebQA as shown in Table 3, highlighting the effectiveness of our method in balancing specialization and generalization.

## H    COMPARISON OF CONTRASTIVE LOGITS CALCULATION IN VCD AND RAGVL

Although both our method and VCD use contrastive logit calculation, there are fundamental differences in their implementation and motivation. Our approach employs contrastive logit calculation during fine-tuning, rather than inference. VCD, by contrast, applies this calculation exclusively during inference and does not involve fine-tuning. Additionally, we introduce two types of noise during training: token-level noise and data-level noise (negatively sampled images). VCD only incorporates token-level noise during inference. By injecting noise at both levels during training, we leverage the $\Delta logits$ as visual correlation weights to reassign the loss for each token, guiding the model to focus on relevant visual elements. Importantly, inference in our method involves standard decoding, not contrastive decoding.

We draw inspiration from VCD, where visual uncertainty amplifies language priors, and contrasting the logits obtained from enhanced language priors with the original logits better highlights visual relevance. However, our motivation extends beyond mitigating irrelevant factors from a single retrieved image to addressing those arising from multiple images. In contrast, VCD focuses on better attending to visual tokens within a single ground truth image. Thus, the motivations underlying our use of contrastive logits differ fundamentally from those of VCD.

## I    MORE CASE STUDIES

In this section, we provide more attention heatmap visualization of cases requiring single image or multiple images for inferencing.

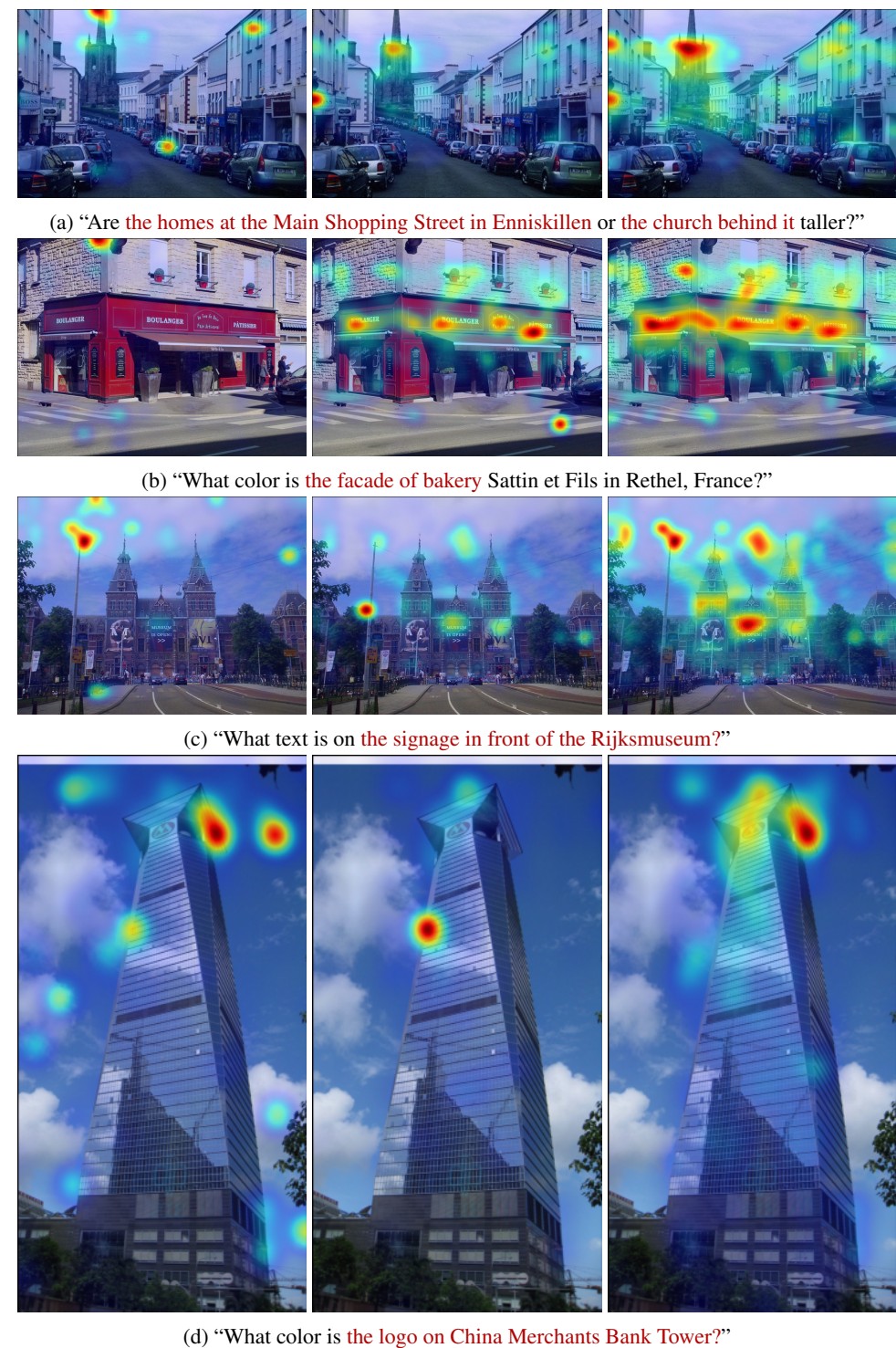

(a) "Are the homes at the Main Shopping Street in Enniskillen or the church behind it taller?"

(b) "What color is the facade of bakery Sattin et Fils in Rethel, France?"

(c) "What text is on the signage in front of the Rijksmuseum?"

(d) "What color is the logo on China Merchants Bank Tower?"

Figure 8: More single-image cases on WebQA.

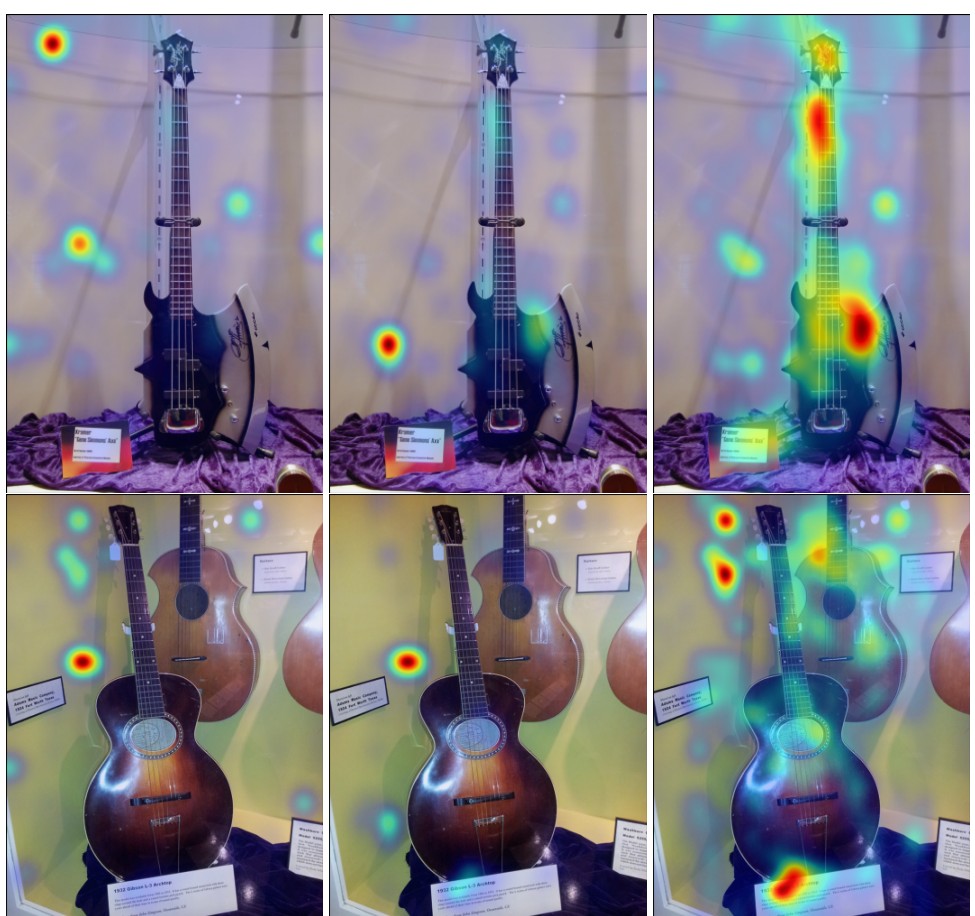

(a) "Which guitar looks more like a tool that might cut a tree; Gene Simmons' Guitar or Gibson L-3 archtop guitar?"

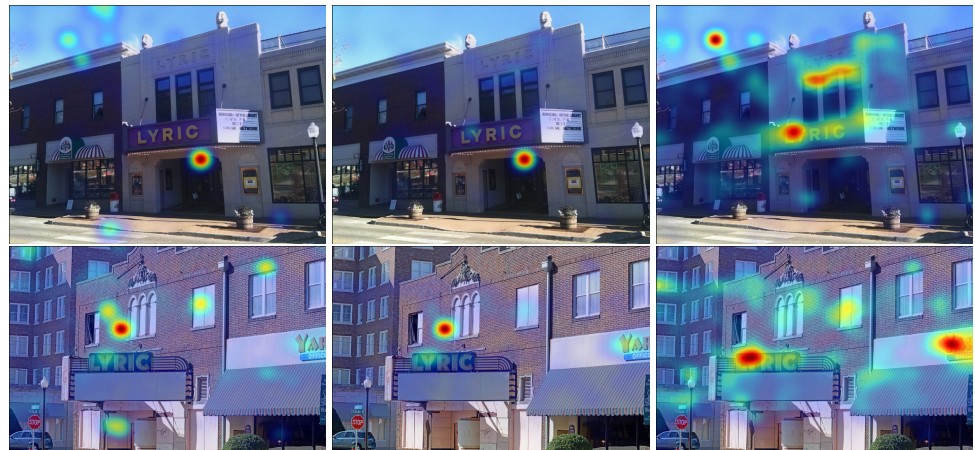

(b) "Are the colors of the word lyric different in the Lyric Theater, Blacksburg and Lyric Theater, Georgia signs?"

Figure 9: More multi-image cases on WebQA.