# OpenReview forum: "MLLM Is a Strong Reranker: Advancing Multimodal Retrieval-augmented Generation via Knowledge-enhanced Reranking and Noise-injected Training"
_ICLR.cc/2025/Conference — Submitted to ICLR 2025_

### Official Review · Reviewer_e7m3 · 2024-10-31

**Soundness:** 2
**Presentation:** 2
**Contribution:** 2
**Rating:** 5
**Confidence:** 4

**Summary:**

The paper aims to address the limitations of MLLMs, which struggle to maintain accurate, up-to-date responses due to their reliance on static training data. To this end, the authors introduce RagVL, a framework that integrates RAG while addressing the multi-granularity noisy correspondence issue. RagVL enhances retrieval performance through knowledge-driven reranking and noise-injected training. The authors employ instruction tuning with a specific template, enabling the model to rank retrieved images effectively and filter top-k candidates for improved retrieval quality. Additionally, noise injection at data and token levels during training further strengthens knowledge-intensive generation. Experiments across four datasets demonstrate RagVL’s effectiveness.

**Strengths:**

+ Study the reranking abilities of MLLMs and integrate RAG for knowledge-intensive VQA.

+ The proposed instruction tuning methods allows for targeted ranking improvements, optimizing retrieval and filtering accuracy.

+ Experimental validation confirms RagVL’s impact on performance across diverse datasets.

**Weaknesses:**

- The authors mentions that they integrate “Multimodal RAG” in the Abstract and Introduction sections, but it seems that the retrieved content is only limited to image modality, so it would be clearer to point out what type of modalities are considered in this work at the beginning.

- The title and abstract are highlighting RAG, but the main content of this manuscript focuses on re-ranking of MLLM, which is inconsistent.

- The motivation of using noise distort image is unclear.

- The authors resort to an existing method, i.e., VCD, for logit contrastive learning and propose to adopt the difference between logits as a visual correlation weight in the MLE loss, but fail to tell clearly why they do that.

- The performance comparison between the proposed method and CLIP seems unfair, since CLIP is not fine-tuned but the proposed method does. To make a fair comparison, it is recommended to fine-tune CLIP using the same training data.

- The description of the adaptive thresholding strategy in Sec. 3.3 is unclear. Besides, the authors “experiment on the validation set and utilize the intersection point ...” in this section, and “report the results on the validation set” in Sec. 4.1. This practice may not be rigorous if the hyperparameters are tuned based on the same set on which the results are reported, which may cause data or label leakage.

- In Tab 2, the two thresholds may perform differently, e.g., adaptive threshold is helpful to Precision but harmful to Recall. More explanations are expected. Besides, how do the two thresholds influence the VQA performance?

- Some unclear experimental details. For example, Do the model “InternVL2-1B w/ CLIP Top-N” undergo next-token prediction fine-tuning with the same training data used in the proposed method? If not, it may be unfair.

**Questions:**

- What does the textual query q denote in Sec. 3.2, questions in VQA scenarios? Please make it clearer.

- Would be the proposed method harmful to the general abilities in some VQA tasks except for knowledge-intensive abilities?

---

> ### Author Response · Authors · 2024-11-15
> **Response to Reviewer e7m3 (1/3)**
>
> Thank you for your constructive comments. We will solve your problems point by point as follows:
> ***
>
> > **W1**: The authors mention that they integrate “Multimodal RAG” in the Abstract and Introduction sections, but it seems that the retrieved content is only limited to image modality, so it would be clearer to point out what type of modalities are considered in this work at the beginning.
>
> **A1**: Figure 1 clearly introduces the motivation and theme of our work, illustrating that the retrieved content consists of best-match image-caption pairs (including both image and text modalities) to ensure robust generation. This highlights that our method explicitly accounts for the relevance between different modalities throughout the “Multimodal RAG” process.
>
>
> > **W2**: The title and abstract are highlighting RAG, but the main content of this manuscript focuses on re-ranking of MLLM, which is inconsistent.
>
> **A2**: Reranking is a key component of the retrieval stage and represents the core contribution of our work. This emphasis is reflected in the main content, as well as prominently in the title, “MLLM is a Strong Reranker”, and the abstract. Additionally, our approach constitutes a comprehensive multimodal RAG method, comprising three stages: retrieval, reranking, and generation. Beyond optimizing the reranking stage, we have also introduced a novel method to enhance the generation stage. To maintain coherence, this narrative is consistently presented in the title and throughout the paper.
>
> The corresponding experimental results for the generation stage are detailed in Table 3, Table 4, and Figure 5, with additional results provided in the Appendix. These results demonstrate the superiority of our proposed method in achieving accurate retrieval and robust generation.
>
> > **W3**: The motivation of using noise distort image is unclear.
> >
> > **W4**: The authors resort to an existing method, i.e., VCD, for logit contrastive learning and propose to adopt the difference between logits as a visual correlation weight in the MLE loss, but fail to tell clearly why they do that.
>
> **A3&4**: As stated in Lines 240-244, we analyze the interleaved challenge in real-world scenarios, which poses the difficulty of determining the optimal number of images to support robust generation. Since it is challenging for the retriever to perform 100% accuracy, irrelevant images included in input prompts would still disturb the generation of MLLM. **Therefore, the model should learn to deal with these MNC (Multi-granularity Noisy Correspondence) situations, where retrieved images and captions might carry irrelevant or noisy information as defined in the Introduction.** That is why we use noise-distorted images during fine-tuning MLLM for more robust generation.
>
> Although both our method and VCD [1] use contrastive logit calculation for a more robust generation of MLLMs, there are fundamental differences in their implementation and motivation. **Our approach employs contrastive logit calculation during training, rather than inference.** VCD, by contrast, applies this calculation exclusively during inference and does not involve fine-tuning. **Additionally, we introduce two types of noise during training: token-level noise and data-level noise (negatively sampled images). VCD only incorporates token-level noise during inference.** By injecting noise at both levels during training, we leverage the $\Delta logits$ as visual correlation weights to reassign the loss for each token, guiding the model to focus on relevant visual elements. Importantly, inference in our method involves standard decoding, not contrastive decoding.
>
> In Lines 245-265, we draw inspiration from VCD, where visual uncertainty amplifies language priors, and contrasting the logits obtained from enhanced language priors with the original logits better highlights visual relevance. **However, our motivation extends beyond mitigating irrelevant factors from a single retrieved image to addressing those arising from multiple images.** In contrast, VCD focuses on better attending to visual tokens within a single ground truth image. Thus, the motivations underlying our use of contrastive logits differ fundamentally from those of VCD.

---

> > ### Author Response · Authors · 2024-11-15
> > **Response to Reviewer e7m3 (2/3)**
> >
> > > **W5**: The performance comparison between the proposed method and CLIP seems unfair, since CLIP is not fine-tuned but the proposed method does. To make a fair comparison, it is recommended to fine-tune CLIP using the same training data.
> >
> > **A5**: We have fine-tuned CLIP to ensure a fair comparison. As shown in Table 1, the method labeled *CLIP-ViT-L/14-336px w/ SFT* represents CLIP fine-tuned on the MultimodalQA and WebQA datasets. Despite this fine-tuning, the performance of CLIP remains subpar.
> >
> > This might be attributed to the nature of the two datasets (WebQA and MultimodalQA), which are designed for multimodal knowledge-seeking question answering. The queries in these datasets often lack sufficient contextual information for CLIP to effectively perform contrastive learning between images and texts, as CLIP is primarily optimized for aligning images with image captions. For instance, the text paired with images in Figure 1 states: “Which is better maintained, the carving on the front of the Palace of the Governor in Uxmal or the Bird carving above the doorway in Mexico, Architecture?” This query provides only the names of the images, offering limited descriptive content.
> >
> > In contrast, our proposed method excels in such scenarios due to the advanced semantic understanding and analytical capabilities of MLLMs, which enable more effective processing of complex multimodal queries.
> >
> > > **W6**: The description of the adaptive thresholding strategy in Sec. 3.3 is unclear. Besides, the authors “experiment on the validation set and utilize the intersection point ...” in this section, and “report the results on the validation set” in Sec. 4.1. This practice may not be rigorous if the hyperparameters are tuned based on the same set on which the results are reported, which may cause data or label leakage.
> >
> > **A6**: The description in Section 4.1 indicates that we use the original validation sets from WebQA and MultimodalQA as test sets in our work. The term “validation set” in Section 3.3, Figure 4, and Figure 7 refers to a newly created subset derived from the original training sets of the benchmark datasets. As such, the training and validation sets in our work are new subsets of the original training data, while the test sets are sourced from the original validation sets. There is no data or label contamination in this setup.
> >
> > We sincerely appreciate your feedback on this point and will ensure greater clarity in future versions of the paper.
> >
> >
> > > **W7**: In Tab 2, the two thresholds may perform differently, e.g., adaptive threshold is helpful to Precision but harmful to Recall. More explanations are expected. Besides, how do the two thresholds influence the VQA performance?
> >
> > **A7**: The proposed thresholds are designed to filter out irrelevant images, preventing the generation model from being misled by irrelevant image inputs and thereby improving precision. However, since the reranker cannot achieve 100% accuracy, some relevant images with low relevance scores may also be filtered out, resulting in a lower recall. As a result, a higher threshold increases precision but reduces recall, and vice versa. In our experiments, adaptive thresholds are often higher than natural thresholds in the same datasets.
> >
> > The adaptive threshold, calculated from the validation set, enhances retrieval accuracy on the test set of a specific dataset. In contrast, the natural threshold is more easily used and can be widely applied across different datasets. Furthermore, our model supports higher recall by lowering the threshold to some extent, as the noise-injected training mitigates the impact of irrelevant images during inference. As shown in Table 3, although the natural threshold might cause a decrease in retrieval accuracy, the QA accuracies obtained from the two thresholds are close.

---

> > > ### Author Response · Authors · 2024-11-15
> > > **Response to Reviewer e7m3 (3/3)**
> > >
> > > > **W8**: Some unclear experimental details. For example, Do the model “InternVL2-1B w/ CLIP Top-N” undergo next-token prediction fine-tuning with the same training data used in the proposed method? If not, it may be unfair.
> > >
> > > **A8**: The model *InternVL2-1B w/ CLIP Top-N* is not fine-tuned, as it is intended for comparison with *RagVL w/o NIT* and *RagVL w/o NIT w/ Natural/Adaptive Threshold* (both not fine-tuned on WebQA or MultimodalQA). This comparison demonstrates that our proposed knowledge-enhanced reranking remains effective even when the models are not fine-tuned. The benefits become even more pronounced after fine-tuning on the two benchmark datasets.
> > >
> > > Additionally, we compared the performance of regular fine-tuning with our method, as shown in Table 4. The row labeled *w/o NLC & ND* represents the model *InternVL2-2B w/ CLIP Top-N* fine-tuned undergoing next-token prediction with the same training data as our proposed method. This approach shows a performance drop compared to our method (*InternVL2-2B w/ CLIP Top-N w/ NLC & ND*), demonstrating the effectiveness of two-level noise injection during training in enhancing the model’s robustness and performance.
> > >
> > > ***
> > > > **Q1**: What does the textual query q denote in Sec. 3.2, questions in VQA scenarios? Please make it clearer.
> > >
> > > **A1**: The textual query q in Section 3.2 denotes the question that needs multimodal knowledge-seeking. For instance in Figure 1, the query is "Which is better maintained, the carving on the front of the Palace of the Governor in Uxmal or the Bird carving above the doorway in Mexico, Architecture?". We would make this clearer in the subsequent version.
> > >
> > > > **Q2**: Would be the proposed method harmful to the general abilities in some VQA tasks except for knowledge-intensive abilities?
> > >
> > > **A2**: Most existing RAG methods primarily focus on knowledge-intensive tasks, even when fine-tuning is involved. For instance, in text-only RAG, Self-RAG[4] has been tested exclusively on knowledge-intensive QA benchmark datasets. Similarly, in multimodal RAG, methods like SURF[2] and MM-Embed[3] are evaluated solely on the test sets of the datasets they were fine-tuned on. These approaches involve additional model training but do not explore tasks beyond the scope of knowledge-intensive applications.
> > >
> > > While training a model on specific tasks can reduce its generalization capabilities (e.g., [5]), a moderate trade-off in universality is often acceptable to significantly enhance task-specific performance. As demonstrated in the table below, we evaluated our approach on three general datasets: MME, MMBench, and SEED-Image. Following noise-injected fine-tuning on WebQA, performance declined only marginally—by 5.2%–5.5%, 1.6%–2.5%, and 1.4%–1.9% on MME, MMBench, and SEED-Image, respectively. However, this fine-tuning resulted in a substantial improvement of approximately 40% on WebQA, highlighting the effectiveness of our method in balancing specialization and generalization.
> > >
> > > | Models | MME | MMBench-EN-test | SEED-Image |
> > > | :-------: | :-----: | :--------------------: | :-------------: |
> > > | InternVL2-1B|1769.2|	61.72|	65.6|
> > > | $\quad$ *w/* WebQA NIT |	1671.3 | 60.76 |	64.32 |
> > > | InternVL2-2B	|1839.8|	72.25|	71.6|
> > > | $\quad$ *w/* WebQA NIT|	1743.2|	70.46|	70.60|
> > >
> > > [1] Leng, S., Zhang, H., Chen, G., Li, X., Lu, S., Miao, C., & Bing, L. (2024). Mitigating object hallucinations in large vision-language models through visual contrastive decoding. In Proceedings of the IEEE/CVF Conference on Computer Vision and Pattern Recognition (pp. 13872-13882).
> > >
> > > [2] Sun, J., Zhang, J., Zhou, Y., Su, Z., Qu, X., & Cheng, Y. (2024). SURf: Teaching Large Vision-Language Models to Selectively Utilize Retrieved Information. arXiv preprint arXiv:2409.14083.
> > >
> > > [3] Lin S C, Lee C, Shoeybi M, et al. MM-Embed: Universal Multimodal Retrieval with Multimodal LLMs[J]. arXiv preprint arXiv:2411.02571, 2024.
> > >
> > > [4] Asai A, Wu Z, Wang Y, et al. Self-rag: Learning to retrieve, generate, and critique through self-reflection[J]. arXiv preprint arXiv:2310.11511, 2023.
> > >
> > > [5] Ling C, Zhao X, Lu J, et al. Domain specialization as the key to make large language models disruptive: A comprehensive survey[J]. arXiv preprint arXiv:2305.18703, 2023.

---

> ### Author Response · Authors · 2024-11-21
>
> Dear Reviewer e7m3,
>
> We wish to express our sincere gratitude for your invaluable feedback! We kindly request that you review our responses and consider updating your assessments accordingly.
>
> We believe that our explanations have thoroughly addressed your concerns. If you have any additional questions or require further clarification, please feel free to contact us. Your final evaluation and potential review update would be greatly appreciated and would significantly contribute to the refinement of our work.
>
> Thank you for your dedication and thoughtful reviews. We look forward to your further response.
>
> Best regards,
>
> Paper#3410 Authors

---

> > ### Comment · Reviewer_e7m3 · 2024-11-25
> > **Response to the Authors**
> >
> > Thanks a lot for your response and care for my comments. The concerns regarding writing and presentation have been addressed and I hope you can further polish these points in the new manuscript to improve the readability. I increase the rating according to your clarification.
> >
> > However, there are still two key concerns on the **novelty** and **significance** of this work:
> >
> > 1. After the careful comparison of this work and [a], I still think there is no substantial technical difference, i.e., Eqn. (5) - Eqn. (7) in NOISE-INJECTED TRAINING are the same as Algorithm 1 in [a].
> >
> > 2. As shown in Tab 4, the improvement of RagVL over RagVL w/o ND & NLC seems not very attractive. Compared with RagVL w/o ND & NLC which uses CLIP-L (less than 0.5B parameters) as the re-ranker, RagVL with InternVL2-2B as the re-ranker may bring much higher computation cost but only gets marginal improvement only on knowledge VQA tasks.
> >
> > [a] Seeing the Image: Prioritizing Visual Correlation by Contrastive Alignment. NeurIPS'24

---

> > > ### Author Response · Authors · 2024-11-25
> > > **Response to Follow-up Questions**
> > >
> > > Thank you for your feedback on our response. Regarding your new questions, our further explanations are as follows:
> > > ***
> > > > **Q1**: After the careful comparison of this work and [a], I still think there is no substantial technical difference, i.e., Eqn. (5) - Eqn. (7) in NOISE-INJECTED TRAINING are the same as Algorithm 1 in [a].
> > >
> > > **A1**:
> > > - In [a], the model is retrained from the pre-training stage with the aim of training a model that focuses more on matching image-text pairs from potentially mismatched datasets. In contrast, our goal is to achieve noise-resistant generation in practical multimodal RAG scenarios. Therefore, we actively injected noise at both the data level and the token level, and we only performed LoRA fine-tuning on knowledge-intensive tasks.
> > > - The logits used for comparison with the original logits in [a] are derived solely from text input, whereas our method utilizes noise-injected images to obtain the logits for comparison.
> > >
> > > Thus, our approach differs from [a] in terms of the way training data is constructed, the actual implementation, and the underlying motivation.
> > >
> > >
> > > > **Q2**: As shown in Tab 4, the improvement of RagVL over RagVL w/o ND & NLC seems not very attractive. Compared with RagVL w/o ND & NLC which uses CLIP-L (less than 0.5B parameters) as the re-ranker, RagVL with InternVL2-2B as the re-ranker may bring much higher computation cost but only gets marginal improvement only on knowledge VQA tasks.
> > >
> > > **A2**:
> > > - In Table 4, the row labeled "w/o ND & NLC" **corresponds to the method that does not use noise-injected training, but still employs InternVL2-2B as the reranker**, ensuring a high recall accuracy and preventing a significant drop in results.
> > > - In contrast, **the row labeled "w/o reranker" only uses CLIP-L for retrieval without employing InternVL2-2B for reranking**, resulting in very low recall accuracy. In comparison, the first row, RagVL ($\eta=0.5$), which uses InternVL2-2B for reranking, achieves a 6.7% improvement in the single image scenario and a 4.1% improvement in the overall scenario, which is not marginal.
> > >
> > > We agree that directly using MLLM for inference reranking is indeed more time-costly compared to CLIP. However, the reranking process can be performed in parallel by scoring multiple images at the same time. We can also reduce the number of images needed to be reranked to decrease the cost of computation and time (in this paper, the number of reranked images per query is 20). Moreover, as mentioned in Appendix C, we can adopt engineering approaches such as FlashAttention[1], PagedAttention[2], and Prompt Cache[3] to pre-cache the attention calculations for repeated prompts or directly include the repeated prompts in the system prompt to reduce inference latency. We believe that with existing methods, it is feasible to achieve significant computational efficiency improvements for the proposed reranking method through engineering optimizations. However, this is not the focus of this paper. In the subsequent version, we will also consider using smaller-parameter MLLMs to reduce this cost further.
> > >
> > > As shown in Table 3, using visual knowledge to assist MLLMs enables more grounded reasoning, which is very important for knowledge-intensive tasks. Models without RAG perform significantly worse on benchmarks compared to models with RAG. Reranking is a crucial step for all RAG systems. For Multimodal RAG, providing MLLM with all top-k images (k=20 for WebQA) would inevitably introduce a large amount of irrelevant noise, which naturally hinders the model’s ability to accurately utilize information from ground truth images, leading to degraded reasoning performance. This highlights the importance of reranking, as failing to rerank often results in irrelevant images being directly retrieved. Therefore, in scenarios where only a few images are selected from retrieved images, improving accurate recall is crucial. As shown in Table 2, on MultimodalQA, our method achieves the same level of recall performance on Recall@1 as CLIP’s Recall@20 (Table 6), illustrating that the reranking step can bring a significant improvement in retrieval accuracy and thus is necessary for Multimodal RAG.
> > >
> > > We hope that our rebuttal addresses your questions. Please let us know if any further questions need clarification.
> > >
> > > [1] Dao T, Fu D, Ermon S, et al. Flashattention: Fast and memory-efficient exact attention with io-awareness[J]. Advances in Neural Information Processing Systems, 2022, 35: 16344-16359.
> > >
> > > [2] Kwon W, Li Z, Zhuang S, et al. Efficient memory management for large language model serving with pagedattention[C]//Proceedings of the 29th Symposium on Operating Systems Principles. 2023: 611-626.
> > >
> > > [3] Gim I, Chen G, Lee S, et al. Prompt cache: Modular attention reuse for low-latency inference[J]. Proceedings of Machine Learning and Systems, 2024, 6: 325-338.

---

> > > > ### Author Response · Authors · 2024-11-28
> > > >
> > > > Dear Reviewer e7m3,
> > > >
> > > > I hope this message finds you well. I am writing to follow up on our latest response to your comments regarding our manuscript. We would like to inquire whether our revisions have adequately addressed your concerns.
> > > >
> > > > We have noticed that your score remains negative and lower than the average scores from other reviewers, and we are eager to understand any additional concerns you may have. Our goal is to enhance the quality of our paper, and your feedback is invaluable in this process.
> > > >
> > > > Thank you for your time and consideration. We look forward to your response.
> > > >
> > > > Best regards,
> > > >
> > > > Paper#3410 Authors

---

### Official Review · Reviewer_DgaV · 2024-11-02

**Soundness:** 3
**Presentation:** 3
**Contribution:** 2
**Rating:** 5
**Confidence:** 5

**Summary:**

In this work, the authors propose the RagVL framework to enhance the performance of Multimodal Large Language Models (MLLMs) in Multimodal Retrieval-Augmented Generation (RAG). The framework filters retrieved images using a fine-tuned MLLM ranker, while a generator model, also fine-tuned on RAG tasks, references these retrieved images to produce improved outputs. During training, the generator is enhanced for robustness through both coarse- and fine-grained noise injection, addressing the challenge of handling incorrect retrievals. Experimental results demonstrate the framework’s effectiveness.

**Strengths:**

Addressing multimodal RAG for MLLMs tackles an important problem.
Experimental findings show the proposed framework’s effectiveness in retrieval tasks and in improving Visual Question Answering (VQA) with RAG.

**Weaknesses:**

The framework may lack substantial novelty. The RagVL approach appears to combine an MLLM ranker with elements from existing techniques, including Visual Contrastive Decoding ([1], for fine-grained noise injection) and Robust RAG Training ([2], for coarse-grained noise injection). Using LLMs/MLLMs as rankers (reward models) in Reinforcement Learning from Human Feedback (RLHF) is already a common approach.


[1]Leng, S., Zhang, H., Chen, G., Li, X., Lu, S., Miao, C., & Bing, L. (2024). Mitigating object hallucinations in large vision-language models through visual contrastive decoding. In Proceedings of the IEEE/CVF Conference on Computer Vision and Pattern Recognition (pp. 13872-13882).
[2]Sun, J., Zhang, J., Zhou, Y., Su, Z., Qu, X., & Cheng, Y. (2024). SURf: Teaching Large Vision-Language Models to Selectively Utilize Retrieved Information. arXiv preprint arXiv:2409.14083.

**Questions:**

1. How does the proposed coarse-grained noise injection differ from and relate to Robust RAG Training?
2. Does noise-injected training affect the MLLM’s general performance (i.e., on standard benchmarks such as MME or MMBench?)

---

> ### Author Response · Authors · 2024-11-15
> **Response to Reviewer DgaV (1/2)**
>
> Thank you for your comments. We will solve your problems point by point as follows:
> ***
>
> > **W1**: The framework may lack substantial novelty. The RagVL approach appears to combine an MLLM ranker with elements from existing techniques, including Visual Contrastive Decoding ([1], for fine-grained noise injection) and Robust RAG Training ([2], for coarse-grained noise injection). Using LLMs/MLLMs as rankers (reward models) in Reinforcement Learning from Human Feedback (RLHF) is already a common approach.
>
> **A1**: As mentioned by Reviewer X8zE in Strength 1:our method "Effectively Mitigating the Multi-Granularity Noisy Correspondence (MNC) Problem: By employing the knowledge-enhanced reranking module, the paper uses instruction tuning to equip MLLMs with stronger semantic understanding and reranking capabilities, enabling more precise selection of visual information relevant to the query. This approach significantly improves retrieval accuracy.". Unlike the reward model in RLHF, which ranks candidate answers generated by LLMs based on human preferences, our reranker is specifically designed for retrieval-augmented generation (RAG). While many studies employ LLMs as rankers in various tasks, such as LLM-as-Judge [6] and RankGPT [7], their primary innovation lies in ranking answers accurately. **What's more, to the best of our knowledge, we are the first to utilize MLLMs as rerankers for multimodal RAG, allowing for effective semantic analysis and improved retrieval.** Subsequent to our work, NVIDIA proposed MM-Embed[3], which finetunes MLLM as universal retrievers and rerankers to tackle diverse tasks, further highlighting the value and innovation of our contribution.
>
> Although both our method and VCD [1] use contrastive logit calculation, there are fundamental differences in their implementation and motivation. **Our approach employs contrastive logit calculation during fine-tuning, rather than inference.** VCD, by contrast, applies this calculation exclusively during inference and does not involve fine-tuning. **Additionally, we introduce two types of noise during training: token-level noise and data-level noise (negatively sampled images). VCD only incorporates token-level noise during inference.** By injecting noise at both levels during training, we leverage the $\Delta logits$ as visual correlation weights to reassign the loss for each token, guiding the model to focus on relevant visual elements. Importantly, inference in our method involves standard decoding, not contrastive decoding.
>
> In Lines 245-265, we draw inspiration from VCD, where visual uncertainty amplifies language priors, and contrasting the logits obtained from enhanced language priors with the original logits better highlights visual relevance. **However, our motivation extends beyond mitigating irrelevant factors from a single retrieved image to addressing those arising from multiple images.** In contrast, VCD focuses on better attending to visual tokens within a single ground truth image. Thus, the motivations underlying our use of contrastive logits differ fundamentally from those of VCD.
>
> At a coarse-grained level, you noted similarities between our method and Surf proposed by Sun [2]. **However, we would like to clarify that our work was publicly released in July 2024 and submitted to ICLR on October 11th, whereas the paper of Surf was first made public on September 21st, 2024. Thus, Surf should be regarded as a concurrent or subsequent work to ours.** Furthermore, there are clear distinctions between our methods. Our approach involves negative sampling of the ground truth image, combining the sampled negatives with context for MLLMs during instruction tuning. During inference, our retriever selects Top-K candidates, which are then reranked into Top-2 (e.g., for WebQA) by the reranker. The noise-injected training enables our model to discern differences among candidates and identify relevant visual elements. Conversely, Sun’s Robust RAG Training emphasizes high-quality dataset construction, where positive and negative labels are pre-generated by MLLMs. During inference, their retriever directly passes Top-K candidates to the generator without reranking.

---

> > ### Author Response · Authors · 2024-11-15
> > **Response to Reviewer DgaV (2/2)**
> >
> > > **Q1**: How does the proposed coarse-grained noise injection differ from and relate to Robust RAG Training?
> >
> > **A1**: **First, we would like to clarify that the publicly released version of our work appeared in July 2024, whereas theirs was first made public in September 2024.** Furthermore, there are clear distinctions between our methods. Our approach involves negative sampling of the ground truth image, combining the sampled negatives with context for MLLMs during instruction tuning. During inference, our retriever selects Top-K candidates, which are then reranked into Top-2 (e.g., for WebQA) by the reranker. The noise-injected training enables our model to discern differences among candidates and identify relevant visual elements. Conversely, Sun’s Robust RAG Training emphasizes high-quality dataset construction, where positive and negative labels are pre-generated by MLLMs. During inference, their retriever directly passes Top-K candidates to the generator without reranking.
> >
> >
> > > **Q2**: Does noise-injected training affect the MLLM’s general performance (i.e., on standard benchmarks such as MME or MMBench?)
> >
> > **A2**: Most existing RAG methods primarily focus on knowledge-intensive tasks, even when fine-tuning is involved. For instance, in text-only RAG, Self-RAG has been tested exclusively on knowledge-intensive QA benchmark datasets. Similarly, in multimodal RAG, methods like SURF and MM-Embed are evaluated solely on the test sets of the datasets they were fine-tuned on. These approaches involve additional model training but do not explore tasks beyond the scope of knowledge-intensive applications.
> >
> > While training a model on specific tasks can reduce its generalization capabilities (e.g., [5]), a moderate trade-off in universality is often acceptable to significantly enhance task-specific performance. As demonstrated in the table below, we evaluated our approach on three general datasets: MME, MMBench, and SEED-Image. Following noise-injected fine-tuning on WebQA, performance declined only marginally—by 5.2%–5.5%, 1.6%–2.5%, and 1.4%–1.9% on MME, MMBench, and SEED-Image, respectively. However, this fine-tuning resulted in a substantial improvement of approximately 40% on WebQA as shown in our paper, highlighting the effectiveness of our method in balancing specialization and generalization.
> >
> >
> > | Models | MME | MMBench-EN-test | SEED-Image |
> > | :-------: | :-----: | :--------------------: | :-------------: |
> > | InternVL2-1B|1769.2|	61.72|	65.6|
> > | $\quad$ *w/* WebQA NIT |	1671.3 | 60.76 |	64.32 |
> > | InternVL2-2B	|1839.8|	72.25|	71.6|
> > | $\quad$ *w/* WebQA NIT|	1743.2|	70.46|	70.60|
> >
> > [1] Leng, S., Zhang, H., Chen, G., Li, X., Lu, S., Miao, C., & Bing, L. (2024). Mitigating object hallucinations in large vision-language models through visual contrastive decoding. In Proceedings of the IEEE/CVF Conference on Computer Vision and Pattern Recognition (pp. 13872-13882).
> >
> > [2] Sun, J., Zhang, J., Zhou, Y., Su, Z., Qu, X., & Cheng, Y. (2024). SURf: Teaching Large Vision-Language Models to Selectively Utilize Retrieved Information. arXiv preprint arXiv:2409.14083.
> >
> > [3] Lin S C, Lee C, Shoeybi M, et al. MM-Embed: Universal Multimodal Retrieval with Multimodal LLMs[J]. arXiv preprint arXiv:2411.02571, 2024.
> >
> > [4] Asai A, Wu Z, Wang Y, et al. Self-rag: Learning to retrieve, generate, and critique through self-reflection[J]. arXiv preprint arXiv:2310.11511, 2023.
> >
> > [5] Ling C, Zhao X, Lu J, et al. Domain specialization as the key to make large language models disruptive: A comprehensive survey[J]. arXiv preprint arXiv:2305.18703, 2023.
> >
> > [6] Zheng L, Chiang W L, Sheng Y, et al. Judging llm-as-a-judge with mt-bench and chatbot arena[J]. Advances in Neural Information Processing Systems, 2023, 36: 46595-46623.
> >
> > [7] Sun W, Yan L, Ma X, et al. Is ChatGPT good at search? investigating large language models as re-ranking agents[J]. arXiv preprint arXiv:2304.09542, 2023.

---

> > > ### Comment · Reviewer_DgaV · 2024-11-17
> > > **Response to Authors**
> > >
> > > Hi, thanks for the explanation and additional experiments. I have adjusted my score accordingly.
> > > Could you further explain the noise contrastive logits part? Why do we want to reweight the token loss according to the contrastive logits (i.e. focusing more on the tokens with a higher sensitivity to the gaussian noise, if my understanding is correct)?

---

> > > > ### Author Response · Authors · 2024-11-18
> > > > **Response to Follow-up Questions**
> > > >
> > > > Thank you for your prompt feedback on our response. Regarding your new question, our further explanation is as follows:
> > > >
> > > > In current vision-language learning, not all fine-grained visual elements are relevant to the paired text. Especially in our Multimodal RAG setting, some of the retrieved images might be unrelated to the query. For instance, in WebQA, some questions only require one single image as supplementary information. However, since it is difficult to predetermine the exact number of ground truth images during the retrieval stage, multiple images are retrieved for reference. In such cases, the generator model needs to learn to focus on the most relevant image during inference while ignoring irrelevant ones (i.e., noise). Therefore, we need a method to highlight the token relevance between vision and text, enabling the model to better attend to relevant visual elements.
> > > >
> > > > VCD analysis suggests that visual uncertainty (enhanced through Gaussian noise) can amplify an MLLM’s dependency on language priors, leading to more severe hallucinations. This results in the generation probability of tokens associated with the image’s visual elements abnormally decreasing. By comparing the model logits before and after enhancing prior dependency, it becomes possible to promote the generation of tokens corresponding to relevant visual elements. Since $\Delta logits$ reflects token relevance, this relevance can also be incorporated into the training process.
> > > >
> > > > Thus, we are not simply focusing more on tokens with a higher sensitivity to Gaussian noise. Instead, by leveraging this relevance-based weighting to re-weight tokens, the model can better focus on relevant visual elements during training, improving vision-language alignment. This, in turn, reduces the impact of noise introduced by retrieval during inference.
> > > >
> > > > We hope that our rebuttal addresses your questions. Please let us know if any further questions need clarification.

---

> > > > > ### Author Response · Authors · 2024-11-28
> > > > >
> > > > > Dear Reviewer DgaV,
> > > > >
> > > > > I hope this message finds you well. I am writing to follow up on our latest response to your comments regarding our manuscript. We would like to inquire whether our revisions have adequately addressed your concerns.
> > > > >
> > > > > We have noticed that your score remains negative and lower than the average scores from other reviewers, and we are eager to understand any additional concerns you may have. Our goal is to enhance the quality of our paper, and your feedback is invaluable in this process.
> > > > >
> > > > > Thank you for your time and consideration. We look forward to your response.
> > > > >
> > > > > Best regards,
> > > > >
> > > > > Paper#3410 Authors

---

### Official Review · Reviewer_ktxp · 2024-11-02

**Soundness:** 3
**Presentation:** 3
**Contribution:** 3
**Rating:** 6
**Confidence:** 2

**Summary:**

This paper proposes RagVL, a novel framework designed to enhance Multimodal Large Language Models (MLLMs) by addressing the multi-granularity noisy correspondence (MNC) problem in multimodal retrieval-augmented generation (RAG). RagVL achieves this through knowledge-enhanced reranking and noise-injected training. By instruction-tuning MLLMs for effective reranking, RagVL improves the accuracy of image-text relevance, allowing models to select top candidates in multimodal retrieval tasks. Additionally, the framework introduces visual noise during training at both data and token levels, bolstering robustness in generation. Extensive experiments on datasets such as WebQA and MultimodalQA demonstrate RagVL's improved retrieval precision and resilience in multimodal question answering, making it effective for scenarios that require fine-grained visual understanding and real-world noise management​.

**Strengths:**

1. This paper is well-motivated.
2. This paper is well-written.
3. The proposed knowledge-enhanced reranking and noise-injected training techniques seem make sense.

**Weaknesses:**

1. Reranking with MLLM is actually very time consuming compared to clip, and MLLM without Captions-aware IT in the Table 1 doesn't gain any performance.
2. The interpretation and analysis of the experimental results are insufficient. For example, in Table 12, why is the effect of no effect ND better in a single image scenario.
3. The performance improvement seems to depend on Image Caption. I wonder if the author's method is limited if no image caption is provided or if the image caption is poor.

**Questions:**

Please refer to the weaknesses.

---

> ### Author Response · Authors · 2024-11-18
> **Response to Reviewer ktxp**
>
> Thank you for your insightful comments. We will solve your problems point by point as follows:
> ***
>
> > **W1**: Reranking with MLLM is actually very time consuming compared to clip, and MLLM without Captions-aware IT in the Table 1 doesn't gain any performance.
>
> **A1**: We agree that directly using MLLM for inference reranking is indeed more time-costly compared to CLIP. However, the reranking process can be performed in a parallel way by scoring multiple images at the same time. We can also reduce the number of images needed to be reranked to decrease the cost of computation and time (in this paper, the number of reranked images per query is 20). Moreover, as mentioned in Appendix C, we can adopt engineering approaches such as FlashAttention[1], PagedAttention[2], and Prompt Cache[3] to pre-cache the attention calculations for repeated prompts or directly include the repeated prompts in the system prompt to reduce inference latency. We believe that with existing methods, it is feasible to achieve significant computational efficiency improvements for the proposed reranking method through engineering optimizations. However, this is not the focus of this paper. In the subsequent version, we will also consider using smaller-parameter MLLMs to further reduce this cost.
>
> As shown in Table 3, using visual knowledge to assist MLLMs in knowledge-intensive tasks enables more grounded reasoning. Models without RAG perform significantly worse on benchmarks compared to models with RAG. Reranking is a crucial step for all RAG systems. For Multimodal RAG, providing MLLM with all top-k images (k=20 for WebQA) would inevitably introduce a large amount of irrelevant noise, which naturally hinders the model’s ability to accurately utilize information from ground truth images, leading to degraded reasoning performance. This highlights the importance of reranking, as failing to rerank often results in irrelevant images being directly retrieved. Therefore, in scenarios where only a few images are selected from retrieved images, improving accurate recall is crucial. As shown in Table 2, on MultimodalQA, our method achieves the same level of recall performance on Recall@1 as CLIP’s Recall@20 (Table 6), illustrating that the reranking step can bring a significant improvement in retrieval accuracy.
>
> Table 1 compares the performance of models without fine-tuning to those fine-tuned using our proposed fine-tuning method for the reranking model. Since MLLMs are typically trained on general datasets, we used a small instruction-tuning dataset to induce the model’s discriminative ability to conduct the reranking task. The results demonstrate that our fine-tuned model can leverage its advanced semantic understanding and analytical capabilities to comprehend queries, making it a powerful reranker. More importantly, as shown in Figure 3, our method is both resource-efficient and highly generalizable, making it easy to apply in any reranking scenario.
>
> > **W2**: The interpretation and analysis of the experimental results are insufficient. For example, in Table 12, why is the effect of no effect ND better in a single image scenario.
>
> **A2**: We found that when using LLaVA as a reranker with a natural threshold for single-image tasks, there is a high probability that only one image is retained after filtering the top-N images (N=2 for WebQA). In this scenario, our noise-injected data struggles to take effect as it aims to make MLLMs focus on the correct image from multiple input images. Thank you for pointing this out. In future versions, we will provide a more in-depth analysis and explanation of this result.
>
> > **W3**: The performance improvement seems to depend on Image Caption. I wonder if the author's method is limited if no image caption is provided or if the image caption is poor.
>
> **A3**: As shown in the example in Figure 1, the captions of the ground truth and the distractors are relatively similar and do not provide additional detailed information (e.g., the details of "the carving on the front" from the query), while similar situations occur frequently in the datasets. However, our method is still able to correctly retrieve both ground truth images, demonstrating that captions and images play a synergistic role in retrieval, both being critically important. This is also why we use MLLM as a reranker, as it can better leverage information from both visual and textual modalities.
>
> [1] Dao T, Fu D, Ermon S, et al. Flashattention: Fast and memory-efficient exact attention with io-awareness[J]. Advances in Neural Information Processing Systems, 2022, 35: 16344-16359.
>
> [2] Kwon W, Li Z, Zhuang S, et al. Efficient memory management for large language model serving with pagedattention[C]//Proceedings of the 29th Symposium on Operating Systems Principles. 2023: 611-626.
>
> [3] Gim I, Chen G, Lee S, et al. Prompt cache: Modular attention reuse for low-latency inference[J]. Proceedings of Machine Learning and Systems, 2024, 6: 325-338.

---

### Official Review · Reviewer_X8zE · 2024-11-04

**Soundness:** 3
**Presentation:** 3
**Contribution:** 3
**Rating:** 8
**Confidence:** 4

**Summary:**

This paper introduces a new framework called RagVL, designed to enhance the performance of Multimodal Large Language Models (MLLMs) in retrieval-augmented generation tasks. The study addresses the common issue of multi-granularity noisy correspondence (MNC) in multimodal retrieval. To tackle this, RagVL employs two main strategies: knowledge-enhanced reranking and noise-injected training. The framework uses instruction tuning to enable the model to accurately select images relevant to the query, and it introduces visual noise during training to improve model robustness. Experimental results demonstrate that RagVL significantly improves retrieval and generation performance across multiple datasets, particularly in scenarios that require fine-grained visual understanding.

**Strengths:**

1. Effectively Mitigating the Multi-Granularity Noisy Correspondence (MNC) Problem: By employing the knowledge-enhanced reranking module, the paper uses instruction tuning to equip MLLMs with stronger semantic understanding and reranking capabilities, enabling more precise selection of visual information relevant to the query. This approach significantly improves retrieval accuracy.

2. Enhanced Model Robustness: Through the noise-injected training method, visual noise is introduced at both the data and token levels, which boosts the model’s robustness when handling multimodal data. Experimental results show that this method notably enhances overall performance in multimodal retrieval and generation tasks, particularly in scenarios that require fine-grained visual understanding.

**Weaknesses:**

1. As presented in Table 2, the two approaches introduced in reranking thresholds show improvements in most metrics but exhibit instability.
2. Although the introduction of the MLLM reranker resulted in some performance improvement, it requires a substantial amount of additional inference time. When using only CIPL to select the top-K, the time taken is 1.23 seconds, but incorporating the reranker adds an extra 5 to 6 seconds.

**Questions:**

1. MLLM possesses strong capabilities, but using it as an additional reranker seems to introduce significant computational overhead. Please discuss scenarios where performance improvements might justify the extra computational cost.
2. The two methods introduced for reranking thresholds have shown improvements across most metrics but also exhibited instability. After incorporating the reranking thresholds, the model’s recall decreased. Please analyze the reasons behind this phenomenon.

---

> ### Author Response · Authors · 2024-11-18
> **Response to Reviewer X8zE**
>
> Thank you for your constructive comments. We will solve your problems point by point as follows:
> ***
> > **W1**: As presented in Table 2, the two approaches introduced in reranking thresholds show improvements in most metrics but exhibit instability.
> >
> > **Q2**: The two methods introduced for reranking thresholds have shown improvements across most metrics but also exhibited instability. After incorporating the reranking thresholds, the model’s recall decreased. Please analyze the reasons behind this phenomenon.
>
> **A1**: The proposed thresholds are designed to filter out irrelevant images, preventing the generator model from being misled by irrelevant image inputs and thereby improving precision. However, since the reranker cannot achieve 100% accuracy, some relevant images with low relevance scores may also be filtered out, resulting in a lower recall but more importantly a significantly higher precision.
>
> We would like to highlight that retrieval accuracy is very important for Multimodal RAG as irrelevant images in the input context will lead to a performance decrease of MLLM during inference. Thus, we adopt two thresholds to filter out images with low confidence after reranking. Although this step might cause a slight decrease in Recall, the removal of irrelevant images brings a positive effect on MLLM reasoning.
>
> > **W2**: Although the introduction of the MLLM reranker resulted in some performance improvement, it requires a substantial amount of additional inference time. When using only CLIP to select the top-K, the time taken is 1.23 seconds, but incorporating the reranker adds an extra 5 to 6 seconds.
> >
> > **Q1**: MLLM possesses strong capabilities, but using it as an additional reranker seems to introduce significant computational overhead. Please discuss scenarios where performance improvements might justify the extra computational cost.
>
> **A2**: We agree that directly using MLLM for inference reranking is indeed more time-costly compared to CLIP. However, the reranking process can be performed in a parallel way by scoring multiple images at the same time. We can also reduce the number of images needed to be reranked to decrease the cost of computation and time (in this paper, the number of reranked images per query is 20). Moreover, as mentioned in Appendix C, we can adopt engineering approaches such as FlashAttention[1], PagedAttention[2], and Prompt Cache[3] to pre-cache the attention calculations for repeated prompts or directly include the repeated prompts in the system prompt to reduce inference latency. We believe that with existing methods, it is feasible to achieve significant computational efficiency improvements for the proposed reranking method through engineering optimizations. However, this is not the focus of this paper. In the subsequent version, we will also consider using smaller-parameter MLLMs to further reduce this cost.
>
> As shown in Table 3, using visual knowledge to assist MLLMs in knowledge-intensive tasks enables more grounded reasoning. Models without RAG perform significantly worse on benchmarks compared to models with RAG. Reranking is a crucial step for all RAG systems. For Multimodal RAG, providing MLLM with all top-k images (k=20 for WebQA) would inevitably introduce a large amount of irrelevant noise, which naturally hinders the model’s ability to accurately utilize information from ground truth images, leading to degraded reasoning performance. This highlights the importance of reranking, as failing to rerank often results in irrelevant images being directly retrieved. Therefore, in scenarios where only a few images are selected from retrieved images, improving accurate recall is crucial. As shown in Table 2, on MultimodalQA, our method achieves the same level of recall performance on Recall@1 as CLIP’s Recall@20 (Table 6), illustrating that the reranking step can bring a significant improvement in retrieval accuracy and thus is necessary for Multimodal RAG.
>
> [1] Dao T, Fu D, Ermon S, et al. Flashattention: Fast and memory-efficient exact attention with io-awareness[J]. Advances in Neural Information Processing Systems, 2022, 35: 16344-16359.
>
> [2] Kwon W, Li Z, Zhuang S, et al. Efficient memory management for large language model serving with pagedattention[C]//Proceedings of the 29th Symposium on Operating Systems Principles. 2023: 611-626.
>
> [3] Gim I, Chen G, Lee S, et al. Prompt cache: Modular attention reuse for low-latency inference[J]. Proceedings of Machine Learning and Systems, 2024, 6: 325-338.

---

> > ### Comment · Reviewer_X8zE · 2024-11-26
> >
> > Thank you for your explanation. I believe it has addressed some of my questions.

---

### Author Response · Authors · 2024-11-25
**General Response**

Dear Reviewers and Meta-Reviewers,

We sincerely thank all the reviewers for their time and their thoughtful comments and questions. To address the concerns, we have submitted a revised version incorporating the following revisions:

- Clarification of the experiment setup. (Section 4.1)
- Evaluation on general benchmark datasets. (Appendix G)
- Comparison of contrastive logits calculation in VCD and RagVL. (Appendix H)

We believe these revisions address the reviewers' concerns and further strengthen our work. Thank you for your constructive feedback and recognition of our contributions.

Best regards,

Paper#3410 Authors

---

### Meta-Review · Area_Chair_ACxV · 2024-12-21

**Metareview:**

This paper introduces a new framework called RagVL, designed to enhance the performance of Multimodal Large Language Models (MLLMs) in retrieval-augmented generation tasks. The study addresses the common issue of multi-granularity noisy correspondence (MNC) in multimodal retrieval. To tackle this, RagVL employs two main strategies: knowledge-enhanced reranking and noise-injected training. The framework uses instruction tuning to enable the model to accurately select images relevant to the query, and it introduces visual noise during training to improve model robustness. Experimental results demonstrate that RagVL significantly improves retrieval and generation performance across multiple datasets, particularly in scenarios that require fine-grained visual understanding.

Strengths:
+ The proposed instruction tuning methods allow for targeted ranking improvements, optimizing retrieval and filtering accuracy.
+ Experimental validation confirms RagVL’s impact on performance across diverse datasets.

Weaknesses:
+ The framework may lack substantial novelty. The RagVL approach appears to combine an MLLM ranker with elements from existing techniques, including Visual Contrastive Decoding.
+ The performance improvement seems to depend on Image Caption

**Additional Comments On Reviewer Discussion:**

After the rebuttal, the submission received mixed reviews (5568). The main remaining concerns are: the novelty and significance of the proposed method are limited. The technical difference with an existing work (NeurIPS'24) is marigal. Overall, I think this submission can be further improved, and recommend Reject.

---

### Decision · Program_Chairs · 2025-01-22

Reject